# Information-based TMS to mid-lateral prefrontal cortex disrupts action goals during emotional processing

R. C. Lapate [1], M. K. Heckner[2], A. T. Phan [3], A. Tambini[4,5] & M. D'Esposito [6]

The ability to respond to emotional events in a context-sensitive and goal-oriented manner is essential for adaptive functioning. In models of behavioral and emotion regulation, the lateral prefrontal cortex (LPFC) is postulated to maintain goal-relevant representations that promote cognitive control, an idea rarely tested with causal inference. Here, we altered mid-LPFC function in healthy individuals using a putatively inhibitory brain stimulation protocol (continuous theta burst; cTBS), followed by fMRI scanning. Participants performed the Affective Go/No-Go task, which requires goal-oriented action during affective processing. We targeted mid-LPFC (vs. a Control site) based on the individualized location of action-goal representations observed during the task. cTBS to mid-LPFC reduced action-goal representations in mid-LPFC and impaired goal-oriented action, particularly during processing of negative emotional cues. During negative-cue processing, cTBS to mid-LPFC reduced functional coupling between mid-LPFC and nodes of the default mode network, including frontopolar cortex—a region thought to modulate LPFC control signals according to internal states. Collectively, these results indicate that mid-LPFC goal-relevant representations play a causal role in governing context-sensitive cognitive control during emotional processing.

Successful cognitive control and goal-directed behavior have long been thought to depend on the function of a frontoparietal network that includes the mid-LPFC, wherein multivariate neural activity patterns have been shown to represent behavioral or task rules[1–10]. Accordingly, individuals with lateral frontal lesions often exhibit behavior that is overly guided by salient external stimuli irrespective of context, termed environmental-dependency syndrome[11,12]. In the domains of emotion and behavioral regulation, function of mid-LPFC is likewise thought to play an important role in instantiating top-down control—LPFC engagement reliably increases during the cognitive regulation of emotion (such as when cognitive reappraisal is used to increase or decrease emotional responses according to one's goals; for

meta-analyses, see[13–15]), and causal perturbations of mid-LPFC typically enhance the context-insensitive influence of emotional stimuli on behavior[16,17] and impair behavior in decision-making tasks requiring self-control[18]. Further implicating intact mid-LPFC function in promoting adaptive emotional responding, individuals with mood and anxiety disorders often show reduced recruitment of mid-LPFC during emotional processing and regulation[19–21], and mid-LPFC lesions are associated with depression risk (to a greater extent than lesions to other PFC regions[22,23]). In emotion regulation models, as in theories of cognitive control, mid-LPFC function is posited to promote goal maintenance and facilitate attentional allocation that benefits goal-directed responses[14,15,24–26].

[1]Department of Psychological & Brain Sciences, University of California, Santa Barbara, Santa Barbara, CA, USA. [2]Institute of Neuroscience and Medicine, Research Centre Jülich, Jülich, Germany. [3]Harvard–MIT Division of Health Sciences and Technology, Cambridge, MA, USA. [4]Center for Biomedical Imaging and Neuromodulation, Nathan S. Kline Institute for Psychiatric Research, Orangeburg, NY, USA. [5]Department of Psychiatry, New York University School of Medicine, New York, NY, USA. [6]Helen Wills Neuroscience Institute and Department of Psychology, University of California, Berkeley, Berkeley, CA, USA. ✉e-mail: lapate@ucsb.edu

However, the causal and specific contributions of LPFC information representation in emotion remain poorly understood. Do goal representations in mid-LPFC support context-sensitive, goal-directed behavior during emotional processing? While this idea is often postulated in theoretical accounts of emotion and behavioral regulation[8,14,24–27], it is rarely tested—doing so would require combining a representational analysis approach with a causal perturbation method[28,29]. Yet, the majority of prior work examining LPFC function in emotional processing in humans uses univariate analyses of correlative fMRI data, which preclude inferences regarding the nature of LPFC representations relevant to the control of emotional behavior (for exceptions, see refs. 8,30). Moreover, evidence for a *causal* role of mid-LPFC function in modulating behavior during emotional processing is lacking[16,17].

Therefore, to test whether mid-LFPC function causally underlies the control of context-sensitive, goal-directed behavior during emotional processing, we combined multivariate pattern analysis of fMRI data with a causal perturbation method—a transcranial magnetic stimulation (TMS) protocol designed to be inhibitory, continuous theta-burst (cTBS)[31]—aimed at temporarily disrupting the representation of action goals in mid-LPFC. We employed an event-related fMRI paradigm requiring goal-directed action during emotional processing—the Affective Go/No-Go (AGNG) task, wherein action goal ("Go" vs. "No-Go") and emotional valence ("Positive" vs. "Negative"; here, happy and fearful faces) are manipulated orthogonally[32,33]. Throughout the task, participants were asked to press a button ("Go") in response to a target emotional facial expression (happy vs. fearful) and to withhold responding ("No-Go") upon the presentation of a nontarget facial expression (happy, fearful, or neutral). Thus, as in a traditional Go/No-Go procedure, the AGNG task requires overriding prepotent responses in "No-Go" trials according to flexible task rules that change over time. Importantly, emotional information informs task goals and interacts with behavioral performance in the AGNG task. When goal-directed action (Go vs. No-Go) and emotion-evoked action tendency (Positive-cue approach vs. Negative-cue avoidance) are congruent, task performance is typically enhanced compared to when they are incongruent

—for instance, negative emotional cues often increase No-Go accuracy compared to positive cues[32,33]. In other words, emotional valence facilitates goal-oriented behavior when it is goal congruent, and hinders it when incongruent. Performance in this task has been shown to be associated with favorable emotion-regulatory outcomes in everyday life, including coping flexibility[34] and depression symptomatology[35,36]. Mid-LPFC, centrally positioned along a putative rostrocaudal axis of cognitive control in LPFC[37], represents action goals in this task[38].

Participants (n = 31) performed the AGNG task in the MRI scanner in 3 sessions conducted on separate days (Fig. 1): First, they underwent a baseline fMRI session (no TMS) where we identified subject-specific multivariate action-goal representations in mid-LPFC used to guide subsequent TMS targeting. Next, participants returned for two TMS +fMRI sessions where cTBS was administered to mid-LPFC vs. a Control site (primary somatosensory cortex/S1), followed immediately by fMRI scanning of the AGNG task (cTBS site order counterbalanced across subjects). We targeted mid-LPFC action goals in an individualized manner, based on individuals' highest multivariate action-goal decoding (i.e., Go vs. No-Go classifier accuracy) within mid-LPFC—an MVPA-based TMS strategy we refer to as information-based TMS[39]. This individualized TMS targeting approach follows prior work[28,39] and aimed to maximize the functional specificity and sensitivity of TMS[28,40,41] in a region characterized by large inter-individual variability in anatomy[42–45] as well as in anatomy-function correspondence[46–48].

The combined information-based TMS+fMRI approach allowed us to determine the impact of a TMS protocol thought to be inhibitory on the strength of goal-relevant mid-LPFC representations and task behavior. Specifically, the putatively disruptive impact of cTBS to mid-LPFC permits inference about causality, which, coupled with the use of multivariate decoding of fMRI data acquired immediately following cTBS, allows for a test of whether cTBS targeting mid-LPFC representations in fact changed its representational content and altered goal-oriented behavior. We hypothesized that cTBS targeting of action-goal representations in mid-LPFC would reduce the strength of action-goal representations in mid-LPFC and impair goal-directed behavior, incurring performance costs in the AGNG task (compared to

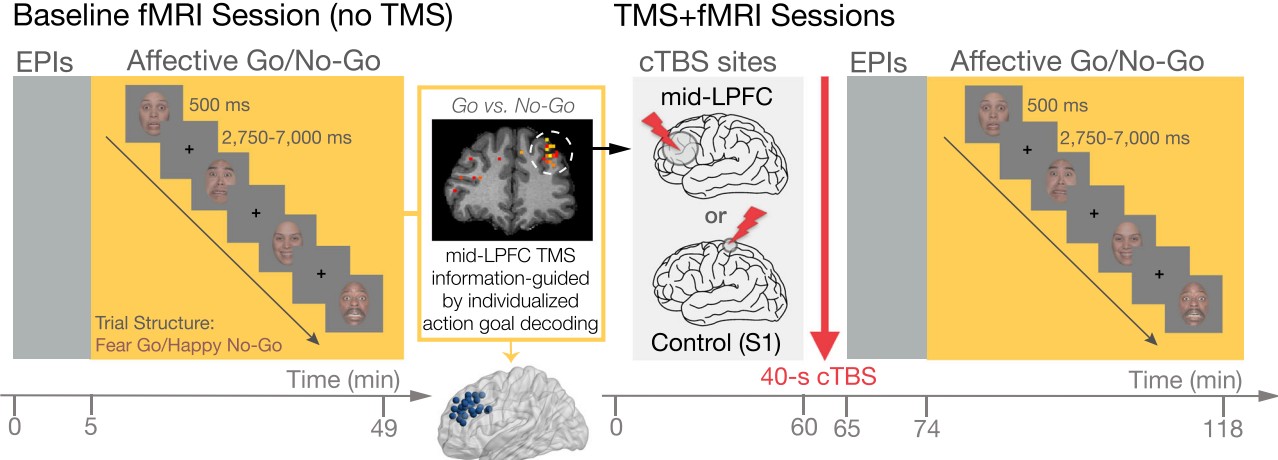

**Fig. 1 | Experimental design and Affective Go/No-Go (AGNG) trial structure.** Participants completed a baseline (no TMS) session and two TMS+fMRI sessions that employed a TMS protocol thought to be inhibitory (continuous theta burst; cTBS): in one session, mid-LPFC was targeted based on the individualized location of multivariate action-goal representations (decoding of "Go" vs "No-Go") as indicated by the results of a searchlight run on fMRI data obtained during the baseline session (data from a representative subject is shown; MNI-transformed coordinates of all subjects (N = 31) are plotted and source data provided as Source Data File). In another session, cTBS was administered to a Control site (S1). Each TMS session was

followed immediately by fMRI scanning. In every session, participants completed the AGNG task, in which they were asked to press a button ("Go") in response to happy or fearful faces, and to withhold responding ("No-Go") following the presentation of nontarget facial expressions. Note that face images shown are not covered by the Creative Commons Attribution license—photographs are from the NimStim Face Stimulus Set. Development of the MacBrain Face Stimulus Set was overseen by Nim Tottenham and supported by the John D. and Catherine T. MacArthur Foundation Research Network on Early Experience and Brain Development. (http://www.macbrain.org/resources.htm.)

baseline (no TMS) and to an active TMS Control site). In addition, we tested whether mid-LPFC cTBS changed mid-LPFC's functional connectivity profile during emotional-cue processing relevant to task goal representations. Optimal behavior in the face of emotional challenges should be sensitive not only to internally maintained, top-down goals, but also to the current emotional context, a process that may rely on medial prefrontal (mPFC) and frontopolar inputs to LPFC[38,49–51]. In summary, we examined the effect of cTBS site (mid-LPFC vs. Control) on the strength of action-goal representations in mid-LPFC, AGNG task performance, and mid-LPFC functional connectivity.

## Results

At baseline, classifier decoding of action-goals in the AGNG task (Go vs. No-Go classifier performance) was above chance in mid-LPFC (anatomical ROI) (AUC $M = 0.533$, $B = 0.033$ (SE = 0.01), $t = 3.215$, Cohen's d = 0.707, $p = 0.007$; for individual-level results, see Supplementary Results and Fig. S1A)[38]. Next, a searchlight analysis of the data obtained during the baseline fMRI session revealed the location of participant-specific peak decoding of action goals within mid-LPFC, which was used for subsequent individualized targeting using cTBS (see Fig. 1 for a representative example and the distribution of mid-LPFC cTBS targets across subjects).

### Information-guided cTBS to mid-LPFC abolishes action-goal representations in this region

Critically, cTBS to mid-LPFC abolished action-goal representations in mid-LPFC, which were no longer decodable in this region (cTBS site main effect $F = 7.117$, $p = 0.0009$; Fig. 2A). After individualized cTBS administration to mid-LPFC, action goal decoding in subject-specific mid-LPFC sites dropped to chance levels (AUC $M = 0.501$, $B = 0.001$ (SE = 0.007), Cohen's d = 0.02, $t = 0.146$, $p$ (vs. chance) = 0.885), and was significantly reduced compared to both baseline and Control (S1) TMS sessions (mid-LPFC vs. baseline: $B = 0.029$ (SE = 0.008), Cohen's d = 0.403, $t = 3.448$, $p = 0.0006$; mid-LPFC vs. Control (S1) TMS: $B = 0.026$ (SE = 0.009), Cohen's d = 0.448, $t = 3.076$, $p = 0.0022$; a non-

parametric analysis replicated these results; Fig. S2; for the cross-subject distribution of mid-LPFC cTBS induced changes in classifier performance relative to baseline and Control TMS sessions, see Fig. S1 C-D). In contrast, after cTBS to the Control site, mid-LPFC action-goal representations remained decodable above chance (AUC $M = 0.527$, $B = 0.027$ (SE = 0.008), Cohen's d = 0.62, $t = 3.518$, $p$ (vs. chance) = 0.0014) to a similar extent that they were at baseline (AUC $M = 0.53$, $B = 0.03$ (SE = 0.011), Cohen's d = 0.526, $t = 2.75$, $p$ (vs. chance) = 0.0137); action-goal decoding in mid-LPFC following Control TMS did not differ from baseline; $B = 0.003$ (SE = 0.008), Cohen's d = 0.023, $t < 0.316$, $p > 0.752$). In summary, cTBS targeting individualized task-relevant action goal signals in mid-LPFC robustly reduced the strength of mid-LPFC action-goal representations (Fig. 2A).

### cTBS to mid-LPFC impairs task performance, particularly during processing of negative cues

Accordingly, TMS targeting mid-LPFC action-goal representations impaired optimal goal-oriented behavior, reducing No-Go accuracy compared to baseline and Control TMS sessions (cTBS site*action goal interaction: $F = 3.124$, $p = 0.044$; mid-LPFC vs. baseline: $B = 5.27$ (SE = 1.29), Cohen's d = 0.519, $t = 4.09$, $p = 0.0001$; mid-LPFC vs. Control (S1) TMS: $B = -2.80$ (SE = 1.27), Cohen's d = 0.231, $t = 2.194$, $p = 0.031$), a finding pronounced during the viewing of negative facial expressions (Fig. 2B). As mentioned, emotional cues often interact with goal-oriented behavior in the AGNG task, facilitating the selection and/or execution of goals congruent with emotional-valence associated actions[32,33,52]. Consistently, at baseline and after Control TMS, fearful faces increased No-Go accuracy compared to happy faces ($ts > 3.53$, $ps < 0.0005$) (Fig. 2B), indicating that unpleasant cues facilitated goal-congruent avoidance in the No-Go condition. However, this goal-congruent facilitation of No-Go performance by negative cues was abolished after mid-LPFC cTBS ($t = 0.529$, $p = 0.597$); cTBS to mid-LPFC significantly reduced No-Go accuracy during negative-cue processing compared to baseline and Control TMS sessions (cTBS site*action goal*valence interaction: $F = 4.133$, $p = 0.016$; mid-LPFC vs. baseline:

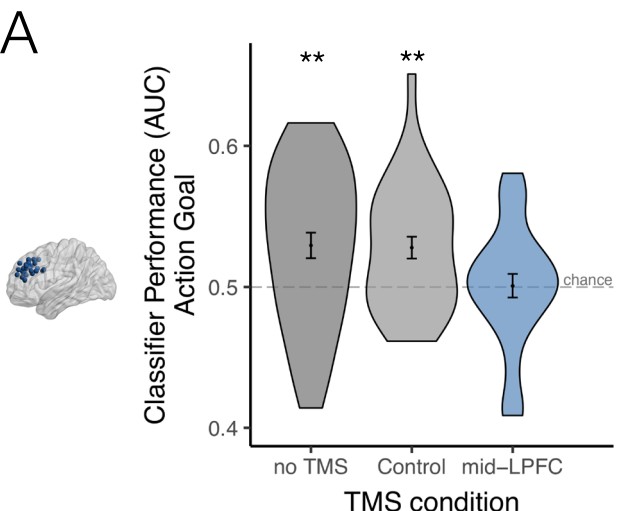

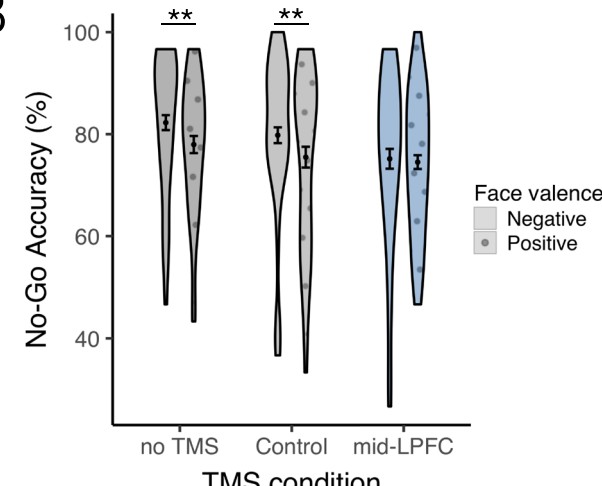

**Fig. 2 | Multivariate classifier performance in LPFC and behavioral results.**
**A** Multivariate classifier performance of action-goal decoding (Go vs. No-Go classifier AUC) from individualized mid-LPFC sites is plotted as a function of TMS condition. Following cTBS to mid-LPFC, action-goal decoding in mid-LPFC dropped to chance, and was significantly reduced compared to both baseline (no TMS) ($t = 3.448$; $p = 0.0006$) and Control (S1) TMS sessions ($t = 3.076$; $p = 0.0022$) (Classifier AUC vs. chance: no TMS: $t = 2.75$, $p_{vs. chance} = 0.0137$; Control TMS: $t = 3.518$, $p_{vs. chance} = 0.0014$; mid-LPFC TMS: $t = 0.146$, $p_{vs. chance} = 0.885$). **B** cTBS to LPFC significantly reduced No-Go performance in the AGNG task (mid-LPFC vs. baseline: $t = 4.09$, $p = 0.0001$; mid-LPFC vs. Control (S1) TMS: $t = 2.194$, $p = 0.031$), abolishing

the increase in No-Go accuracy typically observed during the presentation of negatively-valenced emotional cues (mid-LPFC vs. baseline: $t = 4.660$, $p < 0.0001$; mid-LPFC vs. Control (S1) TMS: $t = 3.060$, $p = 0.0026$). ** Two-tailed $p < 0.0275$. The data of all participants ($N = 31$) are plotted. Colors denote TMS condition: Dark gray = no TMS; Light gray = Control (S1) TMS; Blue = mid-LPFC TMS. Texture denotes emotional-valence condition: Plain = Negative; Circles = Positive. Data are presented as mean values per condition. Error bars: ±1 SEM of the within-subjects difference between conditions[96]. Source data for **A**, **B** are provided as Source Data File.

$B = 7.097$ (SE = 1.523), Cohen's d = 0.622, $t = 4.660$, $p < 0.0001$; mid-LPFC vs. Control TMS: $B = 4.624$ (SE = 1.511), Cohen's d = 0.322, $t = 3.060$, $p = 0.0026$; for the cross-subject distribution of mid-LPFC-TMS induced changes in No-Go Accuracy (Negative – Positive) relative to baseline and Control TMS sessions, see Fig. S1E-F). Importantly, behavioral performance did not differ between baseline and Control TMS sessions ($B = 2.473$ (SE = 1.633), Cohen's d = 0.179, $t = 1.514$, $p = 0.133$). To test whether the observed changes in No-Go accuracy could be explained by more generalized influences of mid-LPFC TMS on comfort and/or motor function, we assessed whether cTBS to mid-LPFC (vs. Control) induced changes in mood and/or response times (RTs); no significant effects were found ($ps > 0.46$; Fig. S3; for additional details and behavioral results, see *Supplementary Information* and Fig. S4). Collectively, these data suggest that mid-LPFC representations play a causal role in promoting goal-directed behavior—here, accurate task performance—in the presence of emotional cues. Further, these results show that the typical potentiation of goal-oriented avoidance by negative cues is reduced when mid-LPFC function is disrupted.

### cTBS targeting of emotion-dependent action goals in mid-LPFC reduces the functional coupling between mid-LPFC and a frontopolar-mPFC network

To further clarify the underpinnings of this reduced potentiation of task-relevant behavior by negative cues after mid-LPFC cTBS, we examined whether cTBS produced circuit-level changes in mid-LPFC's functional connectivity profile during emotional processing. Recent work suggests that cognitive control representations in mid-LPFC may be modulated by emotional cues via interconnected frontopolar cortex[38,49,50,53]. To do so, we used a psychophysiological interaction (PPI) analysis[54] (see *Methods*). We found that during negative emotional processing, cTBS to mid-LPFC reduced the functional coupling between mid-LPFC sites and a frontopolar-mPFC cluster shown in Fig. 3A, comprising frontopolar cortex (BA10) and extending into medial superior frontal gyrus and paracingulate gyrus (hereinafter FP/mPFC) (as well as between mid-LPFC and precuneus and visual cortex,

compared to Control (S1) TMS; whole-brain cluster-corrected for multiple comparisons at $Z < -3.1$, $p < 0.05$; Table S1; results were n.s. during positive emotional processing, in alignment with above-reported behavioral results). The FP/mPFC cluster revealed by this functional connectivity analysis overlapped primarily with the default mode network (DMN)[55] (57.8% of corrected voxels vs. 13.3% with the ventral-attention network; see Tables S2 and S3). Prior work suggests that emotional states and contexts represented in mPFC nodes of the DMN[49,56,57] and frontopolar cortex[38,49] may inform LPFC cognitive control signals via frontopolar projections[49,58]. Accordingly, greater FP/mPFC—mid-LPFC coupling during negative (vs. positive) emotional-cue viewing correlated with stronger action-goal representations remaining in mid-LPFC following mid-LPFC cTBS (Fig. 3B) (mixed-effects model $B = 0.009$, SE = 0.004, $t = 2.01$, $p = 0.046$; analogous results were obtained using an anatomically defined frontopolar ROI, $B = 0.01$, SE = 0.004, $t = 2.57$, $p = 0.011$; Fig. S5). Collectively, these data suggest that inhibitory TMS targeting emotion-dependent action-goal representations in mid-LPFC may disconnect mid-LPFC from an FP/mPFC network that permits affective information to inform goal-directed behavior, potentially via frontopolar projections[38,49,50].

### Control analyses: cTBS targeting of action-goal representations in mid-LPFC does not alter emotional-valence decoding or uni-variate activation in this region

Supporting the functional specificity of our information-based TMS approach, cTBS targeting individualized action-goal representations in mid-LPFC did not alter emotional-valence representations—or the overall magnitude of univariate activation—in subject-specific mid-LPFC sites ($ps > 0.24$) (*Supplementary Information: Control Analyses;* Figs. S6-S7). Moreover, cTBS-induced reductions in action-goal decoding in mid-LPFC were specific to individualized mid-LPFC sites targeted by TMS, and were not observed in *non*-subject specific, anatomically defined mid-LPFC ($p > 0.17$) (Fig. S8) or in *non*-subject specific, functionally defined, mid-LPFC ROIs ($p > 0.214$) (Fig. S9; see *Supplementary Information: Control Analyses: Regional specificity of MVPA changes by mid-LPFC cTBS*).

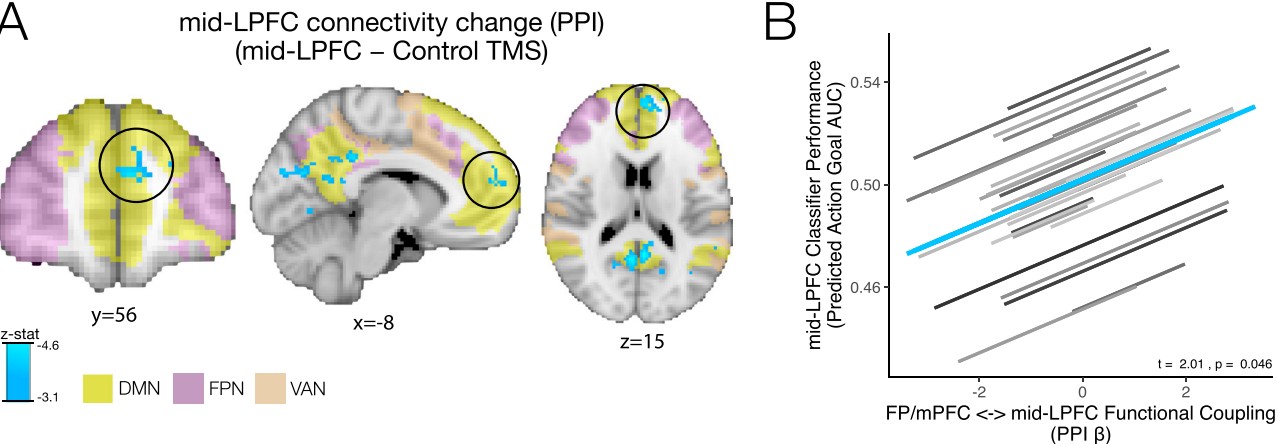

**Fig. 3 | Functional connectivity changes after TMS to mid-LPFC (vs. Control).** The results of a voxel-wise, whole-brain corrected analysis examining changes in mid-LPFC functional connectivity during negative emotional processing following cTBS to mid-LPFC (vs. Control/S1) is shown (PPI analysis cluster-corrected for multiple comparisons, $Z < -3.1$, $p < 0.05$; Table S1). **A** Following cTBS to mid-LPFC, a set of regions that overlapped with the DMN, including a frontopolar-mPFC cluster (referred to as FP/mPFC), showed reduced functional coupling with mid-LPFC sites during negative emotional processing (light blue) compared to after cTBS was administered to the Control site (see Tables S2-S3 for detailed Yeo-7 network overlap). Yeo-7 networks[55]: DMN: Default mode network (yellow). FPN: Fronto-parietal network (magenta). VAN: Ventral attention network (orange). **B** Stronger

functional coupling between mid-LPFC and the FP/mPFC cluster (circled in Fig. 3A) (PPI Beta) was associated with greater mid-LPFC action-goal decoding in the AGNG task as evidenced by a mixed-effects model (across subjects and runs; $B = 0.009$ (SE = 0.004), $t = 2.01$, two-tailed $p = 0.046$), suggesting that emotional information conveyed from frontopolar and mPFC regions may inform action-goal representations in mid-LPFC (analogous results were obtained using an anatomically-defined frontopolar ROI; Figure S5). Mixed-effects model individual-level fits are shown in gray; group-level fit is shown in blue. Source data for **A** (Unthresholded z-stat map) are available at: https://neurovault.org/collections/TSWWVLOM/. Source data for **B** are provided as Source Data File.

## Discussion

In summary, by combining multivariate information-guided TMS with measurements of task-relevant neural activity patterns (fMRI), our results indicate that mid-LPFC function causally modulates goal-directed behavior during emotional processing. Mid-LPFC TMS robustly reduced mid-LPFC action-goal representations and impaired goal-directed behavior in a task requiring cognitive control in response to emotional cues. The combined TMS+fMRI approach allowed us to ascertain the functional specificity of our findings, as cTBS targeting action-goal representations specifically reduced the strength of those representations in individualized mid-LPFC sites without altering emotional valence decoding or average mid-LPFC activity. Therefore, these results help clarify the role of information representation supported by mid-LPFC function during emotional processing, and align with a broader literature underscoring the import of mid-LPFC representations in facilitating successful cognitive control[1–10]. Collectively, these findings are consistent with influential models of behavioral and emotion regulation proposing a central role for mid-LPFC in maintaining goal-relevant representations that promote goal-oriented responding in the face of emotional challenges[14,15,24–26].

While aberrant mid-LPFC function has long been noted in mood and anxiety disorders[20,21,59], whether and how information maintained in LPFC contributes to the control of behavior in emotional contexts has often been hypothesized, but only rarely tested[8,30]. In the future, it will be important to establish whether the nature and function of mid-LPFC goal-relevant representations as revealed by tasks with explicitly cued goals (such as the AGNG task employed in the current study) generalize to spontaneously initiated behavioral goals in less structured scenarios, including abstract goals less coupled to action (motoric) components. Of note, cTBS to mid-LPFC (vs. Control) did not significantly change response times (Fig. S3), suggesting that mid-LPFC action-goal representations targeted with cTBS likely comprised relatively abstract goals that went beyond mere motor preparation and/or execution. Nonetheless, the current dataset does not address whether action-goal states decodable in this region in our study, and altered by mid-LPFC cTBS, pertained to fully abstract action-goal representations versus motoric action plans. Therefore, future studies employing naturalistic paradigms—including self-relevant, ecologically valid stimuli—that capture behavioral goals varying in abstraction, combined with representational analysis approaches[60], will be required to fully unveil the nature and format of LPFC control signals that modulate emotional behavior.

In this study, we used an active control TMS site (somatosensory cortex/S1) that was not expected to influence behavior in the AGNG task. Indeed, performance following Control (S1) TMS was equivalent to that observed at baseline (no TMS session). While the inclusion of an active control site has several advantages relative to using sham stimulation—for instance, permitting more adequate control for non-specific brain tissue changes produced by TMS—stimulation of a posterior site, such as S1, is less likely than mid-LPFC to produce muscle stimulation (i.e., twitching) during cTBS administration that can be unpleasant, even if short lived. Here, as well as in prior work[16], we did not observe differences in mood following cTBS to mid-LPFC compared to S1, suggesting that the potential differences in unpleasantness during cTBS administration to those two sites are unlikely to have produced long-term changes in subjective experience that could confound performance in the AGNG task. Nonetheless, future studies may consider alternative (e.g., lateral frontal) active control TMS sites to better match for potential differences in scalp sensation during TMS administration.

Relatedly, a methodological innovation of the current work was to target mid-LPFC based on the location of individualized action-goal (Go vs. No-Go) multivariate representations, following recent empirical and theoretical work on combined brain stimulation and neuroimaging[28,39] and a prior study showing the increased efficacy of individualized, fMRI-based TMS targeting[40]. Consistent with our core hypothesis, the strength of action-goal decoding in individualized mid-LPFC sites was reduced following cTBS to mid-LPFC (compared to baseline and Control (S1) sessions), an effect that was not observed in anatomical and/or non-individualized mid-LPFC sites (Figs. S8-9). Collectively, these results suggest the import of parsing inter-individual heterogeneity in mid-LPFC function[42–48] and offer support for an individualized, functional-based approach to brain stimulation aimed at understanding causality in brain-behavior relationships[28,39,40], in alignment with a growing emphasis on precision neuroimaging[61,62] and precision psychiatry for brain stimulation[63,64]. Early work systematically comparing individualized vs. group-based TMS targeting approaches demonstrated the increased potency of individualized, fMRI-guided TMS for altering behavior[40], and recent studies of TMS as a treatment for depression have embraced individualized, fMRI-based targeting, serving as the basis for the first FDA-approved individualized fMRI-based TMS protocol for depression treatment[63,65,66]. Note, however, that the current study did not directly test whether individualized targeting of multivariate representations per se was required for the observed findings, as they were examined relative to the active control TMS site (S1) and baseline sessions (rather than relative to a group-coordinate based mid-LPFC TMS site). Therefore, future methodological work systematically quantifying the impact of distinct individualized vs. group-based TMS-targeting strategies (guided by univariate, multivariate, and/or functional-connectivity-based signals) —ideally conducted within subjects—will be required to more fully characterize their differential efficacy in modulating neural activity and behavior, and inform translational neuroscience efforts, including personalized brain stimulation strategies that are increasingly embraced in the clinic[41,63,64].

An influential hierarchical model of the prefrontal organization of cognitive control[67–70] postulates a rostrocaudal gradient of goal abstraction along LPFC, which has been recently revised to highlight a central (apical) role for mid-LPFC function[37,58,71–73]. The frontopolar cortex, once considered the top of this rostrocaudal gradient, has been ascribed a domain-specific role pertaining to the maintenance of temporally extended, internal control signals[58,72]. Here, cTBS to mid-LPFC reduced mid-LPFC functional coupling with a frontopolar-medial PFC region; moreover, stronger frontopolar—mid-LPFC coupling correlated with stronger action-goal representations in mid-LPFC. Therefore, our results affirm a central and causal role for mid-LPFC in orchestrating goal-directed behavior, while also pointing to interactions between mid-LPFC and interconnected frontopolar cortex that may modulate (and be modulated by) cognitive control signals in mid-LPFC as a function of emotional context[27,50,74]. These and other recent findings[38,49–51,53,56] converge to suggest that the frontopolar cortex likely subserves an integrative function spanning beyond temporal or episodic domains to also include emotionally valenced states. These states, which draw on interoceptive and exteroceptive cues, stand as a ubiquitous source of information poised to mobilize control signals for adaptive behavior. Moving forward, uncovering precisely how emotional context and abstract goals become integrated in LPFC to inform behavior will be critical not only for a deeper understanding of mood and anxiety disorders oft-characterized by aberrant mid-LPFC function[19–22,59,75], but also to clarify the organization of cognitive control when challenged by emotionally nuanced scenarios common in everyday life.

## Methods

### Participants

Thirty-one participants were recruited from Berkeley, CA (M = 22.5 y old; SD = 3.32; range = 18–29; 17 female; self-reported data) using UC Berkeley email lists as well as posts in the community (Craigslist). The sample size was chosen based on prior within-subjects TMS studies of prefrontal function and cognitive control (e.g.[37,76]), where we aimed to

maximize initial recruitment due to the number of sessions required to complete the study. Following a baseline fMRI session ($N = 37$), all participants who were invited to participate in the subsequent TMS +fMRI sessions ($N = 31$) returned for all sessions. Eligible participants were healthy, with no self-reported history of psychiatric or neurological disorders, and had normal or corrected-to-normal vision. Written informed consent was obtained from each subject at the University of California, Berkeley. All procedures were approved by the UC Berkeley Committee for the Protection of Human Subjects. Participants were compensated monetarily for their participation. Analysis based on sex or gender were not conducted in this study because we did not have a-priori hypotheses that the effect of mid-LPFC cTBS would vary based on sex or gender.

#### Procedure

**Overview.** Following MRI and TMS safety screening, participants underwent a baseline (no TMS) fMRI session wherein they completed the Affective Go/No-Go task in the MRI scanner. T1-weighted scans were obtained and subsequently used for neuronavigation during the TMS sessions. Participants returned for two additional sessions where offline TMS was administered and followed immediately by completion of the AGNG task inside of the MRI scanner while fMRI data were acquired. As part of a larger study, participants underwent resting state, perfusion, and diffusion-weighted imaging (data not reported here).

The two TMS+fMRI sessions took place on two separate days. Continuous theta-burst TMS (cTBS) was delivered to either mid-LPFC (based on the location of multivariate action-goal representations (Go vs. No-Go decoding) identified in the baseline session), or a control site (medial S1) (see '*TMS sites*' below for details). TMS site order was counterbalanced across participants. The two TMS+fMRI sessions were scheduled as closely as possible based on participants' availability and took place on average 5.63 (SD = 8.24) days apart. Each TMS+fMRI session began with a metal screening, which was followed by a motor thresholding procedure (see '*Transcranial Magnetic Stimulation*' below for details). Before and after the experiment, participants filled out a mood questionnaire (PANAS Now[77]; *Supplementary Information: Control Analysis*).

**Affective Go/No-Go (AGNG) task.** In the MRI scanner in both baseline (no TMS), and after each TMS session, participants completed the AGNG task. This task has good test-retest reliability (see *Supplementary Methods*). Trial design, as well as data pertaining to the baseline (no TMS) session have been reported elsewhere[38]. Briefly, the task comprised six functional runs (~7 min/each). Each run contained 4 action-goal + facial emotion target miniblocks: "*Go Happy, No-Go Fear*", "*Go Fear, No-Go Happy*", "*Go Happy, No-Go Neutral*" and "*Go Fear, No-Go Neutral*". Each miniblock contained 20 trials, 75% (15/20) of which were "Go" trials, and 25% (5/20) were "No-Go" trials. Before each miniblock, participants were instructed to press a button using their right index finger on a handheld button box for faces that matched the "Go" condition, and to withhold pressing the button for faces that matched the "No-Go" condition. Those four miniblocks were presented in counterbalanced orders across the 6 functional runs, and 2 scan run orders were used (counterbalanced across subjects).

In each trial, a fixation cross appeared for 500 ms, followed by a face image that was either a target ("Go") or nontarget ("No-Go") for 500 ms. Then, a 2750–7000 ms inter-trial interval followed (sampled from an exponential distribution). Each AGNG session totaled 80 trials/ run ($n = 480$ trials total across the task) and took ~40 minutes to complete.

**Face stimuli.** Emotional faces (happy, neutral and fearful) consisted of 12 identities (half female) selected from the Macbrain Face Stimulus Set[78] http://www.macbrain.org/resources.htm). Faces were cropped to

remove hair and neck, and matched for average luminance as well as RMS contrast. Faces were presented at 13° x 13° using PsychoPy[79].

**AGNG metrics and behavioral analyses.** As dependent measures, we examined task accuracy and response time (RT). To test whether accuracy and RT were modulated by emotional valence, we used the lme4 package[80], anova and emmeans (https://github.com/rvlenth/ emmeans) functions in R (subject was modeled as a random factor and action-goal, emotional valence and TMS site as random slopes). Effect sizes were computed using the cohensD function from the lsr package.

#### Functional MRI methods

**Image acquisition.** Neuroimaging data were acquired in the UC Berkeley Henry H. Wheeler, Jr. Brain Imaging Center with a Siemens TIM/ Trio 3 T MRI scanner with a 32-channel RF head coil. Whole-brain Blood Oxygen Level-Dependent (BOLD) functional Magnetic Resonance Imaging (fMRI) data were obtained using a T2*-weighted 2x accelerated multiband echo-planar imaging (EPI) sequence (52 axial slices, 2.5 mm³ isotropic voxels; 84 ×84 matrix; TR = 2000 ms; TE = 30.2 ms; flip angle = 80°; 222 image volumes per run). High-resolution T1-weighted MPRAGE gradient-echo sequence images were collected at the end of the session for spatial normalization and TMS neuronavigation (176 ×256 x 256 matrix of 1 mm³ isotropic voxels; TR = 2300 ms; TE = 2.98 ms; flip angle = 9°).

**fMRI data preprocessing.** Functional neuroimaging data were processed using FEAT; FSL version 6.0.1[81,82]. Preprocessing steps included the removal of the first four functional volumes, high-pass filtering (90 s cutoff), FILM correction for autocorrelation in the BOLD signal, slice-time correction, and motion correction using MCFLIRT. Standard and extended motion parameters (i.e., their temporal derivatives and their squares) and a confound matrix containing points of framewise displacement greater than 0.5 mm were used as regressors of non-interest in the analyses to control for movement-confounded activation. Data were smoothed with using a 3 mm full width at half maximum (FWHM) Gaussian filter. Functional images were co-registered to participant's T1-weighted anatomical image using a linear rigid body (6-DOF) transform while maintaining native functional resolution (2.5 mm³ isotropic).

#### Regions of interest (ROIs)

**Anatomical ROIs.** Prefrontal ROIs used for anatomical inference of functional results—frontal pole (FPl & FPm) and mid-LPFC (BA46 & 9-46)—were obtained from the Oxford PFC consensus atlas[83,84], thresholded at 25%, and registered to participants' native surface space using Freesurfer[85]. Vertex coordinates were transformed into the native (volumetric) space, and ROI masks in volumetric space were constructed by projecting half the distance of the cortical thickness at each vertex, requiring that a functional voxel be filled at least 50% by the label, and labeling the intersected voxels[38].

**Functional ROIs.** A 5mm³ ROI surrounding subject-specific mid-LPFC coordinates (see *TMS sites*) was created for the extraction of multivariate action-goal (Go vs. No-Go) decoding as a function of TMS site (no TMS, mid-LPFC and Control/S1).

**fMRI data modeling.** We obtained trial-wise BOLD activation parameters estimates using the Least-Squares All (LS-A) GLM approach[86] and FEAT modeling in FSL[82]. Single trials were modeled using a canonical Double γ hemodynamic response function. For action-goal decoding analyses, we examined the data from correct trials. This amounted to up to 480 trials per session per participant (Mean error = 8.28%, SD = 5.01%). Parameter estimates extracted from each ROI were regularized with multivariate noise normalization[87]. To do so, we obtained an estimate of the noise covariance from the residuals of the

general linear models from each ROI. This matrix was then regularized using the optimal shrinkage parameter, inverted, and multiplied by the vector of betas for each trial[38,87,88]. This approach aims to removes nuisance correlations between voxels that arise due to physiological and instrument noise.

**Multivariate pattern analysis (MVPA).** To test whether cTBS to mid-LPFC altered action-goal representations, we examined classifier performance of Go vs. No-Go classes from multivoxel neural activity patterns extracted from individualized mid-LPFC sites during the baseline session (no TMS), as well as following cTBS to mid-LPFC and Control/S1 sites.

We used a linear classifier to examine the decodability of the action goal (*Go vs. No-Go*) from mid-LPFC multivoxel neural activity patterns implemented with Nilearn[89]. For each subject and ROI, we used a multivariate logistic regression model *(l2 penalty; C = 1)* to iteratively train the classifier on z-scored data. We assessed classifier performance using a leave-one-run-out cross-validation scheme. Classification performance was evaluated using the area under the curve (AUC) metric (i.e., where 0.5 is chance performance). For all multivariate decoding analyses, we used Nilearn's parameter (class_weight = 'balanced') to automatically adjust weights according to class frequencies in the input data (important for the classification of *Go vs. No-Go* classes). To examine whether classifier performance differed from chance, we combined run-wise classifier AUCs (−0.5) across subjects using a mixed-model approach (R package: lme4) and tested whether the intercept differed significantly from 0 (subject and run were entered as random factors). This constituted the primary analysis reported in the manuscript. To test whether classifier performance differed by TMS session, we entered TMS condition as a fixed factor in the same mixed model predicting classifier accuracy (AUC; subject and run were entered as random factors). Effect sizes were computed using the cohensD function from the lsr package in R.

As a control analysis to ascertain the functional specificity of information-based TMS, we extracted neural activity patterns from an anatomically-defined mid-LPFC ROI (see *Anatomical ROIs* above), in which we previously reported that action-goal decoding in the AGNG task is above chance[38] using a larger (fMRI-only) sample comprising the sub-sample examined in the current TMS+fMRI study.

To additionally validate our results using a non-parametric method (i.e., robust to violations of normality assumptions), we derived a null distribution of classifier performance by shuffling labels (*n* = 500 permutations). Results from this non-parametric approach are shown in Fig. S2.

**Functional connectivity analysis.** As described in the main text, to examine whether cTBS to LPFC resulted in circuit-level changes in mid-LPFC connectivity, we used psychophysiological interaction analysis (PPI[54]). To do so, we first extracted the mean time series from individualized mid-LPFC sites. We then ran a separate FEAT analysis for each run that included emotional valence regressors, the demeaned timecourse of participants' mid-LPFC seed, as well as the interaction between this timecourse and regressors for negative and positive emotional stimuli. At the group level, we examined the result of these whole-brain analyses cluster-corrected for multiple comparison at $Z = 3.1$, $p < 0.05$. We examined the % overlap of whole-brain, cluster-corrected results using the Yeo-7 network definition[55]. Given prior findings pointing to frontopolar involvement in the representation of emotional context during the AGNG task[38], we focused the remaining analyses on the frontopolar/mPFC (FP/mPFC) cluster identified by this analysis following cTBS to mid-LPFC. We report the overlap of this FP/mPFC overlap with Yeo 7-network definition and examine whether its functional coupling with mid-LPFC was associated with action-goal decoding in mid-LPFC. To that end, we extracted PPI betas during negative (vs. positive) emotional

processing for each subject and functional run, and entered them into a mixed effects model predicting run-wise Go vs. No-Go classifier accuracy (AUC) in mid-LPFC following cTBS to this region (subject and run were entered as random factors). We repeated this analysis using an anatomically defined frontopolar cortex ROI (see *Regions of Interest-Anatomical ROIs* section above). As potential outliers were observed in the PPI data, run-wise betas that exceeded +/-4 SD from the mean (across subjects) were excluded prior to analyses (1/162 run-wise PPI beta excluded; 0.617%).

## Transcranial magnetic stimulation (TMS)
**TMS sites.** TMS sites targeted in this study were defined on an individualized basis. Left mid-LPFC sites were chosen based on the location of peak multivariate decoding of action goal (Go vs. No-Go) obtained in the baseline AGNG task session as revealed by the result of a spherical searchlight (3-voxel/7.5 mm radius; leave-one run out cross-validation) constrained by a mid-LPFC anatomical mask (defined based on the Oxford PFC consensus atlas as described above; see *ROIs*)[39]. Classifier performance was assigned to each searchlight's central voxel, yielding a map of Go vs. No-Go classifier evidence per participant, which was examined unthresholded in participants' T1-space to select the location of highest evidence of action-goal decoding within the anatomically defined mid-LPFC mask. An example searchlight and MNI-transformed mid-LPFC sites for all participants are shown in Fig. 1; mean MNI-transformed coordinates across participants for mid-LPFC were (x = −35, y = 33, z = 32) with a standard deviation of (x = 8, y = 10, z = 9). Left mid-LPFC (as opposed to right mid-LPFC) was selected for targeting based on prior work detailing the organization of cognitive control in LPFC[37,58], which has revealed more consistent recruitment of left vs. right hemispheres during cognitive control tasks (see[90] for a relevant meta-analysis), as well as our prior work on the causal role of LPFC in emotion, which has likewise revealed more pronounced left (vs. right) LPFC involvement[16,91].

As a control TMS site, we targeted left dorsomedial somatosensory cortex (S1) in a region consistent with the sensory representation of the right foot (approximate MNI coordinate [−10, −38, 78]). The S1 target was located on each subject's native space T1-weighted image based on anatomy. This region was chosen as an active TMS control region due to its circumscribed functional connectivity. Following prior work, targeting a control TMS site permits more rigorous control for non-specific brain tissue effects of stimulation as well as acoustic and scalp sensations compared to a 'no TMS' control condition[16,92–94]. Data pertaining to individualized scalp-to-cortical distances per TMS site, as well as their (non-significant) associations with the magnitude of neural and behavioral effects reported in the current manuscript, can be found in *Supplementary Results* (*TMS sites: scalp-to-cortex distances*).

**TMS stimulation protocol.** TMS was delivered with a Magstim Super Rapid 2 Plus One magnetic stimulator (Magstim, Whitland, UK) using a figure-eight double air stimulating coil with a 70 mm diameter. Precise TMS targeting on a subject-by-subject basis was achieved using a computerized frameless stereotaxic system (Brainsight, Rogue Research) to map the position of the coil and the subject's head in relation to the space of the individual's T1-weighted high-resolution anatomical MRI scan.

To temporarily disrupt the function of mid-LPFC and Control/S1 sites, we used a continuous TMS protocol (cTBS) consisting of 50 Hz trains of 3 TMS pulses repeated every 200 ms continuously over a period of 40 seconds (600 pulses total). This 40-s cTBS protocol has been shown to depress activity in the stimulated brain region for up to 50–60 min after stimulation[31]. Throughout the TMS sessions, experimenters actively maintained the TMS coil in a stable position aided by a MagStim coil holder and continuous monitoring of real-time stereotaxic tracking.

We delivered cTBS at 80% of the active motor threshold, as typically done with this protocol[16,31,92]. Participant's active motor threshold was defined as the lowest stimulus intensity that elicited at least five twitches and/or sensations in 10 consecutive TMS single pulses delivered to the motor cortex while the subject maintained a low-level tonic voluntary contraction of the right first dorsal interosseus. To that end, participants were asked to maintain a pincer grip (using the index and thumb fingers) of about 20% of maximum strength[95]. The active motor threshold across participants varied between 26% to 53% of the maximum stimulator output ($M$ = 38.56%, SD = 5.22%). cTBS was delivered with the coil placed tangentially to the scalp, with the handle pointing posteriorly.

## Reporting summary
Further information on research design is available in the Nature Portfolio Reporting Summary linked to this article.

## Data availability
Study materials, including emotional face stimuli as well as the behavioral and fMRI data, analyzed in this study are available online in OSF at: https://osf.io/3tjsg/. Source data are provided with this paper and available here: https://osf.io/uza3v.

## Code availability
Analyses were run using FSL, Python and R. The code used to run the experiment (Python), conduct multivariate pattern analysis (Python), as well as to reproduce all statistical analysis (R) and graphs (R) reported in the manuscript are available online on OSF at: https://osf.io/3tjsg/.

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

## Acknowledgements

This work was funded by NIH grants MH134000 (R.C.L.), MH063901 (M.D.), F32-MH113347 (R.C.L.), and NSF award BCS-0821855 (M.D.). The authors thank Jacob Miller, Anastasia Kiyonaga, and Jeanette Mumford for helpful discussions, Lennart Verhagen for access to the PFC consensus atlas, Jenna Martin and Jean Wu for assistance with data collection, and Runan Wang for assistance with data processing. Development of the MacBrain Face Stimulus Set, used in this study, was overseen by Nim Tottenham and supported by the John D. and Catherine T. MacArthur Foundation Research Network on Early Experience and Brain Development. (http://www.macbrain.org/resources.htm). Please contact Nim Tottenham at nlt7@columbia.edu for information concerning that stimulus set.

## Author contributions

R.C.L. and M.D. conceptualized and designed the experiment. M.K.H. & A.T. contributed to the experimental design. R.C.L., A.T.P. and M.K.H. collected the data. R.C.L. analyzed the data and wrote the first draft of the manuscript. A.T.P. contributed to the data analysis. All authors contributed to the writing of the final version of the manuscript.

## Competing interests

The authors declare no competing interests.
