## [Peer Review File · Nature Communications]

Information-based TMS to mid-lateral prefrontal cortex disrupts action goals during emotional processingEditorial Note: Parts of this Peer Review File have been redacted as indicated to remove third-party material where no permission to publish could be obtained.

REVIEWER COMMENTS

Reviewer #1 (Remarks to the Author):

This study included an fMRI-guided rTMS session whose effect was measured on behavior as well as fMRI. The study was well justified, and included an active rTMS control condition. The behavioral results were specific to the hypothesized rTMS target and additional mechanistic insight was provided through the addition of the fMRI results.

My primary critique is that the mid-LPFC compared with S1 is a more painful rTMS target which could have presumably interfered specifically with processing the negative cues which is the study's primary behavioral effect.

Also, it is not clear why the behavioral results and fMRI results were not tested for association. This seems to be a missed opportunity.

Distance from scalp to cortex are variable in these targets. Please use the Stokes equation or similar to calculate differences in effective dose of TMS on the per subject and per site level to ensure the primary and control sites were equivalent and to see if there was a relationship between dose and behavioral or fMRI effects.

Please explain why 'sensations' were allowed for detection of an MEP. This is not standard and likely allowed a weaker stimulation level from participants who found TMS to be uncomfortable but not necessarily generating an actual twitch.

Minor

It is too strong of a statement to call the cTBS protocol "inhibitory" given counterexamples in the literature. The strongest statement might be something like "designed to be ..." or "often thought to be..." but generally may be cleaner to not assume this direction.

Could "non-target expression" be operationalized?

Could the searchlight method have missed an even better (decoder accuracy) region?

Please explain how the MVC was determined.

Although some of the TMS choices were not optimal, these can either be explained as caveats or partially mitigated following suggested steps. Overall, this is a valuable contribution to the literature.

Reviewer #2 (Remarks to the Author):

In this manuscript, Lapate and colleagues describe a study combining fMRI and TMS to examine the role of lateral prefrontal cortex in representing action goals during emotional processing. This work is unique in its use of continuous theta burst TMS to target PFC regions involved in cognitive control and assessing concomitant changes in behavior. The use of multivariate approaches to decode action-goal states from PFC activity could provide insights into the effect of TMS that are not possible with other more conventional approaches. There are, however, multiple issues that mitigate my enthusiasm for the paper.

1. Much of the manuscript is based on the idea that prefrontal cortex is particularly important for regulating responses to emotional situations. However, because this work only examines lateral PFC function during the Affective Go/No-Go task, and does not compare the effects of TMS on this task to one that does not involve emotion. For this reason, it is unclear whether the results support the notion that lateral PFC plays a unique role during emotional challenges, or if it functions similarly in emotion vs. non-emotional contexts.

2. Some of the most important findings reported in the manuscript regard action-goal representations in mid-LPFC. The idea of decoding action-goal states and targeting them with TMS is an interesting idea, but multiple aspects of this procedure seem problematic. First, the average performance of within-subject decoding is quite small (mean AUC = .533). This not only limits the possible reduction in decoding performance due to floor effects (a reduction to AUC = .501 with TMS is observed), but it also raises questions as to what exactly is being decoded from LPFC. It seems possible that confounds, rather than representations of action-goals, could be the basis of successful decoding (see, e.g., Todd et al. 2013 Neuroimage).

3. Relatedly, the logic of using decoding to target regions for causal perturbations is not very clear. Regions that contain information about different experimental conditions do not necessarily directly encode variables related to the condition of interest (see, e.g. Haufe et al. 2014 Neuroimage). Because the searchlight localization approach used here is not compared to other methods (e.g., random sampling within the Harvard Oxford Atlas, or using the same coordinate for all participants) it is not clear whether this had any effect on the results.

4. Although facial expressions of fear and happiness are often used to study “emotional processing”, in studying differences in button-pressing to static facial displays, it is not clear whether the Affective Go/No-Go task is capturing the type of emotional events or actions that are alluded to in the abstract and introduction.

Minor comments

1. Showing the classification accuracy for each subject in all three conditions in Figure 2A would help characterize how many subjects showed reductions in decoding performance between noTMS/control and mid-LPFC TMS.

2. How much variability is there in the TMS targeting sites? How much variability would there have been if univariate approaches had been used instead of decoding, or even the center of mass of the target ROI?

Reviewer #3 (Remarks to the Author):

This paper was so fascinating. I would like to highlight the following as particular contributions to the field:

1. Use of representational analysis approach with a causal TMS perturbation method
2. Multivariate action-goal representations in mid-LPFC used to guide individualized TMS targeting
3. The use of an active control site with very interesting decoder results highlighting that this approach dissociated the network effects for the two sites.

This work is highly significant. These are cutting-edge approaches to using TMS-fMRI to causally elucidate action-goal representations. By comparison the bulk of TMS-fMRI studies utilize univariate approaches and do not utilize this elegant task approach coupled with MVPA and importantly concurrent performance data.

It is also noteworthy that this manuscript is very well written, with a strong conceptualization in the extant literature. I found the LPFC DMN conjecture in the discussion to be especially informative. There are likely many clinical implications of this work.

As only minor notes, 1) please expand on the selection of the control site. 2) Please expand on the use of 80% aMT and 600 pulses. It is somewhat surprising that the effects are this strong with such a dose. 3. Please provide test-retest reliability data on the task, especially as the pre to post TMS sessions cannot be contiguous given the MVPA targeting.

REVIEWER COMMENTS

Reviewer #1 (Remarks to the Author):

This study included an fMRI-guided rTMS session whose effect was measured on behavior as well as fMRI. The study was well justified, and included an active rTMS control condition. The behavioral results were specific to the hypothesized rTMS target and additional mechanistic insight was provided through the addition of the fMRI results.

We thank Reviewer 1 for the supportive comments!

My primary critique is that the mid-LPFC compared with S1 is a more painful rTMS target which could have presumably interfered specifically with processing the negative cues which is the study's primary behavioral effect.

We agree with R1 that this is a potential concern given that mid-LPFC TMS is more likely to cause facial muscle contractions relative to the S1 site during TMS administration, and thus be less comfortable. However, we believe this potential short-lived difference in comfort during the stimulation is unlikely to produce long-term changes in negative emotion. As an proxy for this, we did not observe differential TMS-related mood changes following the administration of cTBS to mid-LPFC compared to S1, as assessed by the Positive and Negative Affect Scales (PANAS-Now) (Negative Affect: $t(27) = -0.61$, $p = 0.547$; Positive Affect: $t(27) = -0.748$, $p = 0.461$) (*Supplementary Results: Mood.*)

Following the Reviewer's comment, we additionally examined whether potential TMS-induced changes in unpleasant feelings could be driving the current results. To do so, we tested if the extent to which individuals experienced mood changes following TMS to mid-LPFC (vs. S1) (i.e., Δ Post-Pre_S1 - LPFC across individuals) was associated with changes in task accuracy and/or Reaction Times (RT) in response to fearful (vs. happy) faces across TMS sessions (Δ S1 - LPFC).

Across individuals, the magnitude of TMS-evoked changes in subjective negative and positive mood were unrelated to changes in behavior (all $ps > 0.31$): (Δ S1-LPFC Negative Affect and Δ S1-LPFC No-Go Accuracy (Fear - Happy) $r = 0.009$, $p = 0.96$; Δ S1-LPFC Positive Affect and Δ S1-LPFC No-Go Accuracy (Fear - Happy) $r = 0.063$, $p = 0.76$; Δ S1-LPFC Negative Affect and Δ S1-LPFC RT (Fear - Happy) $r = -0.003$, $p = 0.99$; Δ S1-LPFC Positive Affect and Δ S1-LPFC RT (Fear - Happy) $r = -0.206$, $p = 0.31$). We have now incorporated these additional analyses into *Supplementary Results: Mood* (Page 4). These (null) results are in line with a prior cTBS study by the first author targeting mid-LPFC vs. S1 (control site), in which mood changes by TMS site were likewise not observed, nor did TMS-site-related change scores in mood correlate with task performance—whether mood was measured using the PANAS as in the current study, the STAI (a state anxiety questionnaire), or an implicit mood metric (IPANAT) (Lapate et al. *Psychological Science*, 2017). Thus, while mid-LPFC and S1 sites differ in the extent to which they produce muscle twitches, given the offline nature of the cTBS protocol (40-s administration, followed by MRI scanning after ~5min), we believe changes in comfort are likely to be short-lived: First, they do not seem to produce measurable long-lasting changes in subjective emotional experience—i.e., positive or negative mood. Second, inter-subject variability in TMS-related changes in mood did not correlate with changes in task performance in response to negative cues.

Nonetheless, we fully agree with the Reviewer that an ideal active control site would be matched for scalp sensation and comfort. We added a paragraph to the Discussion section of the revised manuscript to consider this point:

Page 11-12:

In this study, we used an active control TMS site (somatosensory cortex/S1) that was not expected to influence behavior in the AGNG task. Indeed, performance following Control (S1) TMS was equivalent to that observed at baseline (no TMS session). While the inclusion of an active control site has several advantages relative to using sham stimulation—for instance, permitting more adequate control for non-specific brain tissue changes produced by TMS—stimulation of a posterior site, such as S1, is less likely than mid-LPFC to produce muscle stimulation (i.e., twitching) during cTBS administration that can be unpleasant, even if short lived. Here, as well as in prior work (Lapate et al. 2017), we did not observe differences in mood following cTBS to LPFC compared to S1, suggesting that the potential differences in unpleasantness during cTBS administration to those two sites is unlikely to have produced long-term changes in subjective experience that could confound performance in the AGNG task. Nonetheless, future studies may consider alternative (e.g., prefrontal or lateral) active control TMS sites to better match for potential differences in scalp sensation during TMS administration.

Also, it is not clear why the behavioral results and fMRI results were not tested for association. This seems to be a missed opportunity.

We thank Reviewer 1 for raising this point. In light of the sample size of this within-subjects, multi-session TMS+fMRI study (N=31; 3 sessions/subject), we centered our a-priori analyses on group-level main effects and 2-way interactions (as opposed to individual differences analyses). Following the Reviewer's comment, we tested whether the difference (delta) in classifier accuracy following cTBS to LPFC versus S1 (i.e., Δ AUC S1–LPFC for Go/No-Go) was associated with changes in task accuracy in response to negative (vs. positive) cues in the mid-LPFC vs. the S1 cTBS session; Δ S1 – LPFC for each behavioral metric). We did not find that TMS-session differences in classifier accuracy correlated with TMS-session differences in behavior when examining these associations across individuals, $r = -0.067$, $p = 0.72$.

The Reviewer's comment led us to additionally examine the correlation between LPFC Go/No-Go classifier accuracy and No-Go Accuracy (Negative – Positive) at baseline, given the implicit link between these elements in our manuscript; to reiterate, cTBS applied to mid-LPFC resulted in a reduction of both (a) Go/No-Go classifier accuracy and (b) No-Go Accuracy (Fear – Happy). We found that indeed, higher classifier AUC in mid-LPFC tended to correlate with greater No-Go Accuracy (Negative – Positive) at baseline ($r = 0.353$, $p = 0.052$; Figure R1), suggesting the inter-relatedness of the strength of mid-LPFC action-goal decoding and behavioral performance in this task.

It is possible that the above-mentioned delta scores, computed from a single scan in each TMS session (as opposed to from a Post – Pre TMS change in classifier accuracy and behavior *within*

the same day, which would have reduced within-subject variance across sessions, but would have required scanning participants twice on the same day) limited our sensitivity to detect an association between TMS-evoked changes in classifier accuracy and behavior. Given the post-hoc nature of this analysis, and our limited statistical power to detect across-subjects correlations, we opted not to a-priori include these additional analyses in the manuscript (but we are open to doing so if the Reviewer wishes).

Figure R1. The strength of classifier action-goal decoding in mid-LPFC (Go/No-Go) tends to correlate with higher No-Go Accuracy (Negative – Positive) at baseline ($r = 0.353$, $p = 0.052$).

Distance from scalp to cortex are variable in these targets. Please use the Stokes equation or similar to calculate differences in effective dose of TMS on the per subject and per site level to ensure the primary and control sites were equivalent and to see if there was a relationship between dose and behavioral or fMRI effects.

We thank the Reviewer for raising an important methodological question that we had not previously considered.

For the available TMS scalp-to-cortex projection data ($N = 29/31$ subjects) for mid-LPFC and the Control site (S1), Euclidean distances between scalp coordinates and TMS cortical sites were as follows: mid-LPFC $M_{\text{distance}} = 22.13$ mm ($SD = 4.84$); S1 $M_{\text{distance}} = 19.41$ mm ($SD = 2.73$). In other words, mid-LPFC sites were, on average, slightly farther from the scalp than the somatosensory control site ($M_{\text{distance-difference}} = -2.72$ mm, $t(28) = -3.05$, $p = 0.005$). Unsurprisingly, given the individualized mid-LPFC targeting approach, mid-LPFC sites also had more variable distances than S1 (across subjects).

Based on the reviewer's suggestion, we approximated the distance-adjusted motor thresholds (AdjMT) per subject and TMS site, using the average M1 scalp-cortex distance previously reported in Stokes et al. 2005 ($M_{\text{distance}} = 15.1$ mm; $SD = 2$ mm) because precise M1 site scalp-to-cortex projections were not available for this dataset. Following Stokes et al. (2005), we estimated distance-adjusted thresholds using the following formula:

$$\text{AdjMT} = \text{MT} + 3 \times (\text{Distance_TMS_site} - \text{Distance_M1}) \text{ (Stokes et al. 2005).}$$

As expected, given the above-mentioned difference in distances between the TMS sites, adjusted motor thresholds were $M = 59.995$ ($SD = 16.704$) for mid-LPFC and $M = 51.14$ ($SD = 11.943$) for S1, which differed by TMS site, ($t(28) = -3.15$, $p = 0.004$). This indicates that, on average, a higher % of stimulator intensity would be theoretically required to reach mid-LPFC sites with the same intensity relative to the S1 site in our study—although note that there was considerable within-site distance variability in both sites.

We next examined whether TMS-site induced changes in the magnitude of mid-LPFC multivariate decoding as well as No-Go accuracy were associated with individual differences in the distance-adjusted TMS intensities as a function of TMS site ($\Delta AdjMT$ mid-LPFC – S1). We found that the difference in distance-adjusted thresholds between these two cortical sites were uncorrelated with the TMS-induced reduction in mid-LPFC decoding classifier accuracy ($\Delta S1$ –mid-LPFC Classifier AUC scores), ($r(27) = -0.1504$, $p = 0.44$), or with the TMS-induced reduction in No-Go Negative (vs. Positive) task performance accuracy ($\Delta S1$ –mid-LPFC No-Go Accuracy (Negative – Positive), ($r(27) = 0.087$, $p = 0.65$). Thus, despite differences in scalp-to-cortical distances by TMS site (with larger scalp-to-cortex differences observed in mid-LPFC compared to S1), adjusted motor thresholds were unrelated to the magnitude of the observed TMS-induced neural and behavioral effects across subjects.

As a caveat, our present calculation primarily addresses whether relative distances between our two TMS sites differed, as well as the potential behavioral and neural correlates of individual differences in those distances. To estimate the adjusted MT with maximal precision following the Stokes et al. (2005) approach, individualized scalp-cortical distances in M1 data would be required, as the cortical depth of the hand area varies across subjects (Stokes et al. 2005). Because we believe this is an important and oft-neglected point in the existing TMS literature, we added the following section to Supplementary to serve as a reference for future readers.

Supplementary Information (P. 10-11):

TMS sites: scalp-cortex distances

Following an anonymous Reviewer suggestion, and in light of the fact that distances from scalp to cortex can vary across individuals and TMS targets, we estimated scalp-to-cortex distances for mid-LPFC and S1 TMS sites to assess whether this may account for inter-subject variability of the influence of TMS. To do so, we calculated the approximate difference in the effective dose of TMS per subject and TMS site using the Stokes et al. (2005) equation¹⁹ and examined associations between the estimated distance-adjusted TMS intensity (per site) and the neural and behavioral effects reported in this study.

The Euclidean distances between scalp coordinates and TMS cortical locations for the available TMS projection data ($N = 29/31$ subjects) for mid-LPFC and the Control site (S1) were as follows: mid-LPFC $M_{distance} = 22.133$ mm ($SD = 4.838$ mm); S1 $M_{distance} = 19.411$ mm ($SD = 2.734$ mm). Thus, mid-LPFC sites were, on average, located further from the surface ($M_{distance-difference} = -2.72$ mm, $t(28) = -3.05$, $p = 0.005$) than S1 sites. To approximate the adjusted motor threshold (AdjMT) per subject and TMS site, we used a previously-reported average M1 scalp-cortical distance estimate ($M_{distance} = 15.1$ mm; $SD = 2$ mm)¹⁹, as precise M1 projections were not available for this study. Following Stokes et al. (2005), we estimated distance-adjusted motor thresholds using the following formula for each TMS site:

$$AdjMT = MT + 3 \times (Distance_TMS_site - Distance_M1)$$

(where AdjMT is the distance-adjusted MT in percentage stimulator output; MT is each participants' motor threshold; Distance_TMS_site is the scalp-to-cortex distance per TMS site, and Distance_M1 is the scalp-to-cortex distance for the M1 site)

We found that distance-adjusted motor thresholds differed by TMS site, ($t(28) = -3.15, p = 0.004$); such that mid-LPFC's was higher ($M = 59.995$ ($SD = 16.704$)) than S1's ($M = 51.14$ ($SD = 11.943$)). This indicates that, on average, a higher % of stimulator intensity would be theoretically required to target mid-LPFC sites in our study with the same intensity as S1—although note that there was considerable variability across subjects for both sites.

We next examined whether TMS-site induced changes in the magnitude of mid-LPFC multivariate decoding as well as changes in No-Go accuracy were associated with individual differences in the distance-adjusted TMS intensities as a function of TMS site (Δ AdjMT DLPFC – S1). We found that differences in the distance-adjusted thresholds between TMS sites were unrelated to the TMS-induced reduction in mid-LPFC (vs. S1) decoding accuracy (Δ S1–mid-LPFC Classifier AUC scores), ($r(27) = -0.1504, p = 0.44$), or to the TMS-induced reduction in No-Go Negative (vs. Positive) task performance (Δ S1–mid-LPFC No-Go Accuracy (Negative – Positive), ($r(27) = 0.087, p = 0.65$).

Thus, while TMS scalp-to-cortical distances differed by TMS (with larger scalp-to-cortex differences observed in mid-LPFC compared to S1), distance-adjusted motor thresholds were unrelated to the magnitude of the observed TMS-induced neural and behavioral effects (across subjects). However, note that individualized scalp-to-cortex M1 distances were not available for this study—therefore, our present calculation primarily addresses whether relative distances between our two TMS sites differed, as well as the potential behavioral and neural correlates of individual differences in those distances. To estimate the adjusted MT with maximal precision, given that the cortical depth of the hand area can vary across subjects¹⁹, future studies should obtain individualized scalp-cortical distances in M1.

Please explain why 'sensations' were allowed for detection of an MEP. This is not standard and likely allowed a weaker stimulation level from participants who found TMS to be uncomfortable but not necessarily generating an actual twitch.

We thank the Reviewer for inviting us to clarify this point. Since EMG recordings were not available when the data for this study were collected, we relied on visual inspection of muscle twitches to determine the motor threshold (Rossi et al. 2009). However, since visual inspection tends to lead to higher estimated thresholds compared to EMG-based methods (Westin et al. 2014), there is the possibility of estimating higher stimulation intensities than intended. To counteract this potential risk, we also considered subjectively-reported sensations as an indicator of motor activation to potentially help protect against overestimation of one's motor threshold. Despite allowing for sensations as an indication of motor activation, note that a twitch was observed prior to or in conjunction with the report of a sensation in the majority of cases (> 90%).

Importantly, we obtained the motor threshold independently in the two TMS+fMRI sessions—immediately preceding each cTBS administration—which allowed us to examine the test-retest reliability of the motor thresholds obtained using our method. To do so, we computed the Pearson's correlation between the motor thresholds obtained in the first and second TMS session (TMS site—LPFC vs. S1—was counterbalanced across sessions).

We found that the motor thresholds had good test-retest reliability across the two TMS sessions, $r = 0.88$, $p < 5.2e^{-11}$, $95\%CI = [0.768 \ 0.942]$:

Minor

It is too strong of a statement to call the cTBS protocol “inhibitory” given counterexamples in the literature. The strongest statement might be something like “designed to be ...” or “often thought to be...” but generally may be cleaner to not assume this direction.

We thank the Reviewer for this important suggestion. We have replaced instances of “inhibitory TMS/cTBS” with “putatively inhibitory TMS” and/or “a TMS protocol thought to be inhibitory”.

Abstract (P. 2):

“Here, we altered mid-LPFC function in healthy individuals using a **putatively** inhibitory brain stimulation (cTBS) protocol, followed by fMRI scanning.”

Introduction (P. 4):

“(…) we combined multivariate pattern analysis of fMRI data with a causal perturbation method—**a TMS protocol designed to be inhibitory**, continuous theta-burst (cTBS) (Huang et al. 2005) (…)”

Introduction (P. 5):

“The combined information-based TMS+fMRI approach allowed us to determine the impact of a TMS protocol **thought to be inhibitory** on the strength of goal-relevant mid-LPFC representations and task behavior.”

Figure 1 Legend (P. 5):

“Participants completed a baseline (no TMS) session and two TMS+fMRI sessions that employed a TMS protocol **thought to be inhibitory** (continuous theta burst; cTBS) (…)”

Results (P. 9):

“Collectively, these data suggest that **putatively** inhibitory TMS targeting emotion-dependent action-goal representations in mid-LPFC may disconnect mid-LPFC from an FP/mPFC network that permits affective information to inform goal-directed behavior (...)”

Could “non-target expression” be operationalized?

Yes, we expanded this section to clarify target vs. nontarget facial expressions:

Introduction (P. 4):

“Throughout the task, participants were asked to press a button (“Go”) in response to a **target** emotional facial expression (**happy vs. fearful**) and to withhold responding (“No-Go”) upon the presentation of a nontarget **facial** expression (**happy, fearful, or neutral**).”

Could the searchlight method have missed an even better (decoder accuracy) region?

We thank Reviewer 1 for this question. Our a-priori hypotheses centered on the role of mid-LPFC action-goal representations on the modulation of emotional behavior—thus, the searchlight results were examined in conjunction with (i.e., overlaid on) an a-priori mid-LPFC ROI in each participants’ native space.

In response to the Reviewer’s question, we aggregated the results of whole-brain searchlights across subjects to visualize them in MNI (group) space (see Figure R2 below). For reference, a volumetric, group-level approximation of the mid-LPFC ROI is shown in blue. Bilateral decoding accuracy estimated to be above chance is seen in mid-LPFC, in addition to the following other cortical regions: insula, supplementary motor area (SMA), somatomotor cortex (lateral aspect), paracingulate gyrus, and superior parietal lobule. Subcortical regions included bilateral thalamus and left putamen.

Amongst those regions, numerically higher decoding accuracy (compared to mid-LPFC) was seen in the insula, SMA, superior parietal lobule and somatomotor cortex, in alignment with the known involvement of these regions in cognitive control and/or action planning. However, note that we hesitate to (over)interpret these relative regional differences in decoding accuracy as regions outside of PFC are known to have a higher decoding base rate (compared to PFC) due to intrinsic differences in functional SNR as well as regional differences in the local clustering of functionally-specific neurons (Bhandari, Gagne, and Badre 2018) (*see also our response to Reviewer 2 Question #2*); thereby compromising their direct comparability.

Figure R2: Group-level searchlight results (Classifier Accuracy) aggregated across subjects are shown in MNI space.

Please explain how the MVC was determined.

To obtain the active motor threshold, we instructed participants to maintain a voluntary low-level tonic contraction of the right first dorsal interosseus (i.e., while maintaining a pincer grip using the index and thumb fingers) of about 20% of maximum strength (Groppa et al. 2012). The active motor threshold across participants varied between 26% to 53% of the maximum stimulator output ($M = 38.56\%$, $SD = 5.22\%$).

We now clarify this section of the paper:

Methods: TMS stimulation protocol (P. 18):

Participant's active motor threshold was defined as the lowest stimulus intensity that elicited at least five twitches and/or sensations in 10 consecutive TMS single pulses delivered to the motor cortex while the subject maintained a low-level tonic voluntary contraction of the right first dorsal interosseus. To that end, participants were asked to maintain a pincer grip (using the index and thumb fingers) of about 20% of maximum strength⁸⁹. The active motor threshold across participants varied between 26% to 53% of the maximum stimulator output ($M = 38.56\%$, $SD = 5.22\%$). cTBS was delivered with the coil placed tangentially to the scalp, with the handle pointing posteriorly.

Although some of the TMS choices were not optimal, these can either be explained as caveats or partially mitigated following suggested steps. Overall, this is a valuable contribution to the literature.

We thank Reviewer 1 again for the great questions and supportive words!

Reviewer #2 (Remarks to the Author):

In this manuscript, Lapate and colleagues describe a study combining fMRI and TMS to examine the role of lateral prefrontal cortex in representing action goals during emotional processing. This work is unique in its use of continuous theta burst TMS to target PFC regions involved in cognitive control and assessing concomitant changes in behavior. The use of multivariate approaches to decode action-goal states from PFC activity could provide insights into the effect of TMS that are not possible with other more conventional approaches. There are, however, multiple issues that mitigate my enthusiasm for the paper.

1. Much of the manuscript is based on the idea that prefrontal cortex is particularly important for regulating responses to emotional situations. However, because this work only examines lateral PFC function during the Affective Go/No-Go task, and does not compare the effects of TMS on this task to one that does not involve emotion. For this reason, it is unclear whether the results support the notion that lateral PFC plays a unique role during emotional challenges, or if it functions similarly in emotion vs. non-emotional contexts.

We thank the Reviewer for bringing up this important point that should be clarified in the manuscript. First, we did not intend to imply that LPFC plays a *unique* role in governing emotional responses. Rather, we see the role of LPFC in domain-general cognitive control (in non-emotional contexts) as relatively well established, e.g., as mentioned in our Introduction (Page 3):

“Successful cognitive control and goal-directed behavior have long been thought to depend on function of a frontoparietal network that includes the mid-LPFC, wherein multivariate neural activity patterns have been shown to represent behavioral or task rules (Cole, Ito, and Braver 2016; Waskom et al. 2014; Rigotti et al. 2013; Mante et al. 2013; E. K. Miller and Cohen 2001; Duncan 2010; Sakai and Passingham 2006; Morawetz et al. 2016; Jackson et al. 2021; Menon and D’Esposito 2022). Accordingly, individuals with lateral frontal lesions often exhibit behavior that is overly guided by salient external stimuli irrespective of context, termed environmental-dependency syndrome (Knight and D’Esposito 2003; Fuster 2001).”

In contrast, there is a paucity of data pertaining to the causal role(s) of mid-LPFC function in emotion. While theoretical models of behavioral and emotion regulation often postulate that mid-LPFC function supports representations that promote context-sensitive, goal-directed behavior during emotional processing, this idea is only very rarely tested—and that is what we sought to do in the current study.

The Reviewer’s comment made us realize that while we emphasized the well-known role of LPFC in cognitive control in the Introduction, we did not revisit it in the Discussion—which is important to accurately contextualize the “take homes” from our study. Thus, we revised the Discussion section in our manuscript to clarify this important point:

Discussion (P. 10-11):

Therefore, these results help clarify the role of information representation supported by mid-LPFC function during emotional processing, and align with a broader literature underscoring the import of mid-LPFC representations in facilitating successful cognitive control (Cole, Ito, and Braver 2016; Waskom et al. 2014; Rigotti et al. 2013; Mante et al. 2013; E. K. Miller and Cohen 2001;

Duncan 2010; Sakai and Passingham 2006; Morawetz et al. 2016; Jackson et al. 2021; Menon and D'Esposito 2022).

We additionally relate the findings from the current manuscript to the broader body of literature on cognitive control in the last paragraph, as follows:

Discussion (P. 12):

“An influential hierarchical model of the prefrontal organization of cognitive control (Badre and D’Esposito 2009; Koechlin, Ody, and Kouneiher 2003; Badre 2008; Koechlin and Summerfield 2007) postulates a rostro-caudal gradient of goal abstraction along LPFC, which has been recently revised to highlight a central (apical) role for mid-LPFC function (Nee and D’Esposito 2016, 2017; Nee 2021; Badre and Nee 2018; Pitts and Nee 2022). The frontopolar cortex, once considered the top of this rostro-caudal gradient, has been ascribed a domain-specific role pertaining to the maintenance of temporally extended, internal control signals (Badre and Nee 2018; Nee 2021). Here, cTBS to mid-LPFC reduced mid-LPFC functional coupling with a frontopolar-medial PFC region; moreover, stronger frontopolar—mid-LPFC coupling correlated with stronger action-goal representations in mid-LPFC. **Therefore, our results affirm a central and causal role for mid-LPFC in orchestrating goal-directed behavior, while also pointing to interactions between mid-LPFC and interconnected frontopolar cortex that may modulate (and be modulated by) cognitive control signals in mid-LPFC as a function of emotional context (Peters and D’Esposito 2016; Hare, Camerer, and Rangel 2009; Bramson et al. 2020). These and other recent findings (Lapate et al. 2022; Dixon et al. 2018; Satpute and Lindquist 2019; Koch et al. 2018; Bramson et al. 2020; Folloni et al. 2019) converge to suggest that frontopolar cortex likely subserves an integrative function spanning beyond temporal or episodic domains to also include emotionally valenced states. *These states, which draw on interoceptive and exteroceptive cues, stand as a ubiquitous source of information poised to mobilize control signals for adaptive behavior.*”**

2. Some of the most important findings reported in the manuscript regard action-goal representations in mid-LPFC. The idea of decoding action-goal states and targeting them with TMS is an interesting idea, but multiple aspects of this procedure seem problematic. First, the average performance of within-subject decoding is quite small (mean AUC = .533). This not only limits the possible reduction in decoding performance due to floor effects (a reduction to AUC = .501 with TMS is observed), but it also raises questions as to what exactly is being decoded from IPFC. It seems possible that confounds, rather than representations of action-goals, could be the basis of successful decoding (see, e.g., Todd et al. 2013 Neuroimage).

We appreciate Reviewer 2’s point, which resonates with extant methodological work highlighting that decoding accuracies in PFC are notoriously lower than in other brain regions. As recently reviewed in a comprehensive meta-analysis by Bhandari et al. 2018, for two-way classifications, decoding accuracies in PFC often hover between 53-55% [e.g. (Nelissen et al. 2013; Woolgar et al. 2011; Bode and Haynes 2009)]—consistent with the mean AUC = .533 decoding accuracy obtained in the baseline fMRI session in the current study. Bhandari et al. found that the average base rate of decoding in PFC across $n = 76$ studies was 55.4%, which was significantly lower than what is typical in visual (66.6%) and temporal (71.0%) cortices (Bhandari, Gagne, and Badre 2018; Coutanche, Solomon, and Thompson-Schill 2016). This is the case even for features such as task rules, which are known to be robustly represented in the PFC of non-human primates (Bhandari, Gagne, and Badre 2018). That said, Bhandari et al. emphasize that decoding

accuracies do not represent effect sizes—for decoding accuracies to be interpretable, the underlying variance must be taken into account (P. 1474): “*Low decoding accuracies can nevertheless be reliably different from chance (even reflecting a large effect), and therefore be meaningfully interpreted*”.

That was indeed the case in our study: while the overall action-goal decoding values might appear numerically small, an examination of effect sizes reveals that they were moderate in size in our sample. Action goal in the baseline session (vs. chance): Cohen’s $d = 0.707$; Control (S1) TMS (vs. chance): $d = 0.655$. In both cases, those values indicate moderate effect sizes, thereby reducing the floor effect concern that Reviewer 2 raised.

Regarding the reliability of these effects, it is worth noting that TMS to mid-LPFC produced highly consistent effect sizes when compared to *both* baseline and the TMS Control (S1) sessions (Cohen’s $d = 0.403$ and $d = 0.448$, respectively). We attribute the reliability of these effects to our high number of trials per subject and TMS site ($n=440$ /session, obtained over 6 experimental runs; $n=1320$ trials/subject total). Indeed, obtaining classifier decoding accuracies in PFC above chance levels, given the expected base rates, requires sufficient within-subjects data (Bhandari, Gagne, and Badre 2018). We have now included those effect sizes in the revised manuscript (P. 6-7).

With that as context, Bhandari et al. argue that studies with decoding accuracies in PFC *outside* that typical range (e.g., higher than 57.0% — the upper 95% CI of PFC base rate decoding in this meta-analysis) should ‘merit extra attention’ and raise suspicion of potential confounds. In their investigation, they found that the majority of the time, these higher-than-usual decoding accuracies were driven by univariate differences related to general attentional or task engagement—aligning with the concern raised in Todd et al. 2013 that the Reviewer pointed out (Todd, Nystrom, and Cohen 2013).

Lastly, we believe that the finding indicating that TMS to mid-LPFC only changed decoding accuracy in the individualized functional ROI targeted based on its multivariate decoding (Fig. 2A)—but not in the larger, non-individualized anatomical mid-LPFC ROI (Fig. S8)—provides additional evidence for the functional specificity of our TMS results, minimizing the concern that decoding accuracy may have been reduced due to a non-specific task confound.

3. Relatedly, the logic of using decoding to target regions for causal perturbations is not very clear. Regions that contain information about different experimental conditions do not necessarily directly encode variables related to the condition of interest (see, e.g. Haufe et al. 2014 Neuroimage). Because the searchlight localization approach used here is not compared to other methods (e.g., random sampling within the Harvard Oxford Atlas, or using the same coordinate for all participants) it is not clear whether this had any effect on the results.

We thank the Reviewer for raising this important point: While fMRI-based TMS targeting has been advocated for some time [e.g. (Sack et al. 2009)], individualized targeting approaches are often defined using functional connectivity and/or univariate-based metrics [e.g., (Cash, Weigand, et al. 2021; Wang et al. 2014; Freedberg et al. 2019; Tambini and D’Esposito 2020; Tambini, Nee, and D’Esposito 2018; Klooster et al. 2022)] rather than multivariate decoding. Thus, our approach, while not entirely new [see e.g.: (Rose et al. 2016; Polanía, Nitsche, and Ruff 2018)], is relatively novel, compared to other fMRI targeting approaches. Below, we detail our rationale for (a) employing individualized, fMRI-based TMS targeting more generally, as well as (b) why we chose

the multivariate approach. Finally, we address (c) the potential concern related to using backward models as detailed in Haufe et al. 2014 (Haufe et al. 2014).

First, the reason we opted to do individualized, fMRI-based TMS targeting (as opposed to using a group fMRI or MRI coordinate) is because this approach has been shown to be more sensitive and statistically powerful when compared to targeting based on a group-fMRI or MRI-based coordinate (Sack et al. 2009). While results are usually similar in terms of directionality, Sack et al. found that the effect sizes of parietal TMS to alter behavior progressively *increased* when employing individualized fMRI coordinates (Cohen's $d = 1.13$; $n=5$ subjects required) compared to a group fMRI-based coordinate (Cohen's $d = 0.82$; $n=9$ subjects required) or MRI-based targeting (Cohen's $d = 0.67$; $n=13$ subjects required); statistical power was weakest, as expected, when using an EEG-based TMS coordinate (Cohen's $d = 0.34$; $n=47$ subjects required). This has motivated us to conduct much of our prior work using fMRI-based TMS targeting, including in mid-LPFC (e.g., (Lapate et al. 2017; Nee and D'Esposito 2017)). Across various bodies of literature, individualized, fMRI-based TMS has been shown to be effective for altering memory circuitry function in our and others' prior work (Tambini and D'Esposito 2020; Freedberg et al. 2019; Wang et al. 2014); and efficacious (relative to group maps) in protocols for DLPFC-TMS treatment of depression (Klooster et al. 2022; Cash, Weigand, et al. 2021; Cash, Cocchi, et al. 2021).

We believe that using an individualized, fMRI-based TMS targeting—*versus* a group-based, even if functionally derived, coordinate—is particularly crucial when targeting LPFC sites, which have some of the highest anatomical heterogeneity across individuals (J. A. Miller et al. 2021; Rajkowska and Goldman-Rakic 1995; Gordon et al. 2015; Fischl et al. 2008) as well as the highest variability in anatomic-functional correspondence across individuals (Paquola et al. 2019; Vázquez-Rodríguez et al. 2019; Mueller et al. 2013). That is, the location of a group result, particularly in LPFC, may not correspond to the location of any one individual subject's native-space result. Thus, while we have used group-based fMRI coordinates in our prior LPFC work (Lapate et al. 2017, 2020), we believe that individualized fMRI targeting may be particularly important, a-priori, to maximize functional specificity and sensitivity when targeting LPFC.

The multivariate-based targeting approach used here followed directly from Rose et al., *Science*, 2016 (Rose et al. 2016)—which was highlighted in a recent review of non-invasive brain stimulation in humans as one of the suggested multimodal fMRI-TMS pipelines to establish brain-behavioral relationships (Polanía, Nitsche, and Ruff 2018) (**Fig 5A**):

[REDACTED]

While we use offline TMS+ fMRI (*as opposed to concurrent TMS+EEG*) during the task, the core underlying aim is similar: to maximize TMS target specificity and to objectively quantify the extent to which the intended 'representation'/information (in both cases, as evidenced by a classifier decoding performance) was altered by TMS. Indeed, while we do not want to use circular reasoning, we note that this action-goal decoding based TMS approach altered (a) the magnitude of action-goal (but not valence) decoding in mid-LPFC, and (b) a key behavior of interest (No-Go accuracy)—thereby suggesting a reasonable level of psychological specificity achieved by the

current approach. Nonetheless, in alignment with the Reviewer's point, we believe that future work systematically comparing the efficacy of individualized (including multivariate-based) TMS sites vs. group- coordinate based TMS targeting is important; in many cases, the required within-subjects manipulation (e.g., individualized vs. group or anatomical site targeted in the same individuals) and explicit statistical comparison (such a test of the interaction of the efficacy of TMS by TMS approach type) is lacking.

We have now added this point to the manuscript (on the section that compares results obtained with individualized vs. anatomically defined mid-LPFC ROI):

Supplementary Information (P. 10):

Moving forward, future work systematically comparing the efficacy of individualized TMS sites (compared to anatomical or group-coordinate based TMS targeting), ideally conducted within subjects, will be important to further quantify and establish the magnitude of the hypothesized advantages (as well as potential drawbacks) associated with each of those distinct TMS-targeting strategies.

Finally, we appreciate the Reviewer's concern regarding the interpretability of the weights in a backward model (Haufe et al. 2014), although we believe that Haufe et al. partially addresses the Reviewer's concern (P. 106; underline added):

[REDACTED]

In other words, according to the authors, our searchlight approach, which was further constrained by an anatomically defined ROI (encompassing mid-LPFC: BA9-46 & 46—wherein action rules are known to be represented in non-human primates) does not a-priori suffer from the inherent drawbacks of interpreting the weights of a *whole-brain* backward multivariate model.

Given the relative novelty of our approach, we agree with the Reviewer that it is important to provide further information about why we employed it. Thus, we now further elaborate on our rationale in the revised manuscript:

Introduction (P. 5):

This individualized TMS targeting approach follows prior work (Rose et al. 2016; Polanía, Nitsche, and Ruff 2018) and aimed to maximize the functional specificity and sensitivity of TMS (Sack et

al. 2009; Cash, Cocchi, et al. 2021; Polanía, Nitsche, and Ruff 2018) in a region characterized by large inter-individual variability in anatomy (J. A. Miller et al. 2021; Rajkowska and Goldman-Rakic 1995; Gordon et al. 2015; Fischl et al. 2008) as well as in anatomy-function correspondence (Paquola et al. 2019; Vázquez-Rodríguez et al. 2019; Mueller et al. 2013).

4. Although facial expressions of fear and happiness are often used to study “emotional processing”, in studying differences in button-pressing to static facial displays, it is not clear whether the Affective Go/No-Go task is capturing the type of emotional events or actions that are alluded to in the abstract and introduction.

We thank the Reviewer for inviting us to clarify the significance of this work for the types of context-appropriate emotional responding processes that are mentioned in the introduction of our manuscript.

First, we fully appreciate the importance of naturalistic emotional stimuli (for instance, containing self-relevant stimuli) in emotion regulation research. In the context of this methodologically novel (re: TMS+fMRI) study, we chose to adopt a paradigm that had been previously validated—including in prior work indicating that it can capture the more complex types of emotion-regulatory processes that impact everyday life. Specifically, responses in the AGNG task, despite often employing facial expressions as the evocative stimuli (as pointed out by the Reviewer), have been shown to relate to emotional functioning in the following ways. First, individuals with depression are often slower to respond to happy vs. sad stimuli (Erickson et al. 2005; Harfmann, Rhyner, and Ingram 2019). Moreover, depressed adolescents as well as adults with a history of suicide attempt make more commission errors (i.e. false alarms) when viewing negative (vs. positive) no-go stimuli (Harfmann, Rhyner, and Ingram 2019; Kilford et al. 2015). This is worth noting given the involvement of LPFC hypoactivation in depression, which is detectable both during passive negative stimulus viewing as well as during cognitive control tasks (Pizzagalli and Roberts 2022). Using a joystick-based approach/avoidance task similar to the AGNG, Roelofs’ group has also demonstrated that (reduced) frontopolar and dorsolateral PFC engagement during this task is prospectively associated with PTSD symptom development in trauma-exposed police recruits as well as with the magnitude of stress reactivity during social stressors (produced by performing mental arithmetic in front of a panel of judges) (Kaldewaij et al. 2021, 2019). Finally, a recent study examined associations between performance in the AGNG task and self-reported measures of voluntary emotion regulation as well as coping flexibility in response to aversive and potentially traumatic events (Myruski et al. 2017). The authors found that coping flexibility was associated with lower *false alarms to fearful faces*—i.e., which we produced in our study following putative mid-LPFC inhibition via cTBS (Myruski et al. 2017). Of note, the Myruski et al. study included ERP measurements, and showed that the magnitude of the N2 component in response to fear (vs. happy) faces, a prefrontal ERP component typically associated with successful No-Go inhibition, was linked to lower anxiety and depression symptoms (Myruski et al. 2017).

These findings obtained with emotional Go/NoGo (and related paradigms) are not surprising if one views emotion-regulatory processes from an action-control perspective—a (relatively) novel view of emotion regulation that is receiving growing attention (Roelofs, Bramson, and Toni 2023; Bramson, Toni, and Roelofs 2023) and that we embrace in our ongoing work.

In sum, we concur with the Reviewer regarding the importance of clarifying the rationale for the selection of this task in the current manuscript, as well as with the inclusion of more naturalistic

stimuli in future work. To that end, we address these points in the Introduction and Discussion of the revised manuscript:

Introduction (P. 4):

Performance in this task has been shown to be associated with favorable emotion-regulatory outcomes in everyday life, including coping flexibility (Myruski et al. 2017) and depression symptomatology (Harfmann, Rhyner, and Ingram 2019; Kilford et al. 2015).

Discussion (P. 11-12):

While aberrant mid-LPFC function has long been noted in mood and anxiety disorders^{8,30}, **whether and** how information maintained in LPFC contributes to the control of behavior in emotional contexts has often been hypothesized, but only rarely tested^{60,61}. In the future, it will be important to establish whether the nature and function of mid-LPFC goal-relevant representations as revealed by tasks with explicitly cued goals (such as the AGNG task employed in the current study) generalize to spontaneously initiated behavioral goals in less structured scenarios, including abstract goals **less coupled to** action (motoric) components. Of note, cTBS to mid-LPFC (vs. Control) did not significantly change response times (**Figure S3**), suggesting that mid-LPFC action-goal representations targeted with cTBS likely comprised relatively abstract goals that went beyond mere motor preparation and/or execution. Future studies employing naturalistic paradigms—including **self-relevant, ecologically valid stimuli**—that capture behavioral goals varying in abstraction, combined with representational analysis approaches^{63–66}, will be required to **fully** unveil the nature and format of LPFC control signals that modulate emotional behavior.

Minor comments

1. *Showing the classification accuracy for each subject in all three conditions in Figure 2A would help characterize how many subjects showed reductions in decoding performance between noTMS/control and mid-LPFC TMS.*

We thank Reviewer 2 for this helpful suggestion. We found that 21/31 (67.74%) subjects showed decoding performance reductions following mid-LPFC TMS relative to the baseline (no TMS) session, and 20/31 subjects (64.52%) showed a decoding reduction relative to the active Control (S1) session. Following the Reviewer's suggestion of showing subject-level data, we opted to plot these data as the classifier AUC difference score (delta) between condition pairs (mid-LPFC versus baseline and Control (S1)) for ease of visualization of the respective distributions of TMS-induced reductions in classifier decoding across subjects, as follows:

Supplementary Results (P. 3):

Figure S2. (A & B) Subject-level change scores observed in mid-LPFC classifier accuracy (AUC) for action goal decoding (Go/No-Go) following mid-LPFC cTBS are shown relative to **(A)** the TMS Control (S1) fMRI session and **(B)** the baseline (no TMS) fMRI session.

This figure is now included in the revised manuscript (**Figure S2**).

P. 7:

“(...) for the cross-subject distribution of mid-LPFC cTBS induced changes in classifier AUC relative to baseline and Control TMS sessions, see **Figure S2 A-B**).

Relatedly, we have now included analogous plots for the behavioral data (No-Go accuracy) following mid-LPFC cTBS (vs. baseline and Control TMS) in the revised manuscript (**Fig. S2 C & D**):

Figure S2. (C & D) Subject-level change scores observed in No-Go accuracy (Negative – Positive) in the AGNG task following mid-LPFC cTBS are shown relative to **(A)** the TMS Control (S1) session and **(B)** the baseline (no TMS) session.

P. 8:

“(…) for the cross-subject distribution of mid-LPFC cTBS induced changes in No-Go Accuracy (Negative – Positive) relative to baseline and Control TMS sessions, see **Figure S2 C-D**).

2. *How much variability is there in the TMS targeting sites? How much variability would there have been if univariate approaches had been used instead of decoding, or even the center of mass of the target ROI?*

As briefly mentioned in the *Methods*, the average MNI-transformed coordinates across participants for the mid-LPFC TMS site based on individualized (G/NG) MVPA decoding were [$x = -35$, $y = 33$, $z = 32$] with a standard deviation of [$x = 8$, $y = 10$, $z = 9$] (see Fig. 1 for the location of TMS targets across participants). Following the Reviewer’s question, we obtained the average location and across-subject variability for the corresponding univariate contrasts. The MNI coordinates for the peak univariate responses in mid-LPFC were as follows: [Go – No-Go] contrast: $M = [x = -26, y = 42, z = 27]$ with a similar standard deviation as the multivariate site: $SD = [x = 8, y = 10, z = 17]$; for the [No-Go – Go] contrast: $M = [x = -35, y = 44, z = 20]$, $SD = [x = 5, y = 12, z = 14]$.

Thus, the MVPA-based site was ~1cm more posterior and ~0.5-1cm more dorsal in mid-LPFC compared to the peak Go/No-Go univariate contrasts. The variability of the peak across subjects is similar when comparing MVPA vs. univariate metrics—with slightly greater variability observed along the ventral-dorsal (z) axis for the univariate (vs. MVPA) approach.

Reviewer #3 (Remarks to the Author):

This paper was so fascinating. I would like to highlight the following as particular contributions to the field:

- 1. Use of representational analysis approach with a causal TMS perturbation method*
- 2. Multivariate action-goal representations in mid-LPFC used to guide individualized TMS targeting*
- 3. The use of an active control site with very interesting decoder results highlighting that this approach dissociated the network effects for the two sites.*

This work is highly significant. These are cutting-edge approaches to using TMS-fMRI to causally elucidate action-goal representations. By comparison the bulk of TMS-fMRI studies utilize univariate approaches and do not utilize this elegant task approach coupled with MVPA and importantly concurrent performance data.

It is also noteworthy that this manuscript is very well written, with a strong conceptualization in the extant literature. I found the LPFC DMN conjecture in the discussion to be especially informative. There are likely many clinical implications of this work.

First, we thank Reviewer 3 for the enthusiastic and supportive words!

As only minor notes

- 1) please expand on the selection of the control site.*

We chose to target the left dorsomedial somatosensory cortex (S1) as an active TMS control site, which was located in each subject's native space T1-weighted image based on anatomy (i.e., by locating the central sulcus and selecting the most medial and superior aspect of the post-central gyrus) on a subject-by-subject basis (*Methods: P. 18*). Targeting an active control site permits more rigorous control for non-specific brain tissue effects of stimulation, as well as acoustic and general scalp sensations, compared to a 'no TMS'/sham control condition. This approach followed from prior TMS investigations of the causal contributions of prefrontal function in other domains (Lapate et al. 2020, 2017; Hamidi et al. 2009; Tambini and D'Esposito 2020). As mentioned in the *Methods* ('TMS sites'), the S1 site we used is consistent with the location of the representation of the right foot in human subjects (approximate MNI coordinate [-10, -38, 78])—and we chose it not only because its representational function was a-priori unrelated to the task employed in this experiment, but also because of its relatively local and circumscribed functional connectivity with the rest of the brain (Yeo et al. 2011):

Neurosynth derived resting-state functional connectivity map for the TMS control site/medial S1 thresholded at $r > 0.4$ (N = 1,000 subjects).

From: http://neurosynth.org/locations/-10_-38_78_6/

[REDACTED]

While we believe that altering neural excitability in this area likely minimized interference with a cognitive control task employing facial emotional stimuli while controlling for non-specific effects of cortical stimulation, in our revised manuscript, we also acknowledge Reviewer 1's concern that stimulation of S1 is less likely to produce unpleasant facial muscle twitches during stimulation compared to a prefrontal site. Thus, in our revised Discussion, we encourage investigators to potentially consider alternative prefrontal (or more lateral) stimulation sites as active control sites in future studies:

Discussion (P. 11):

In this study, we used an active control TMS site (somatosensory cortex/S1) that was not expected to influence behavior in the AGNG task. Indeed, performance following Control (S1) TMS was equivalent to that observed at baseline (no TMS session). While the inclusion of an active control site has several advantages relative to using sham stimulation—for instance, permitting more adequate control for non-specific brain tissue changes produced by TMS—stimulation of a posterior site, such as S1, is less likely than mid-LPFC to produce muscle stimulation (i.e., twitching) during cTBS administration that can be unpleasant, even if short lived. Here, as well as in prior work (Lapate et al. 2017), we did not observe differences in mood following cTBS to LPFC compared to S1, suggesting that the potential differences in unpleasantness during cTBS administration to those two sites is unlikely to have produced long-term changes in subjective experience that could confound performance in the AGNG task. Nonetheless, future studies may consider alternative (e.g., prefrontal or lateral) active control TMS sites to better match for potential differences in scalp sensation during TMS administration.

2) Please expand on the use of 80% aMT and 600 pulses. It is somewhat surprising that the effects are this strong with such a dose.

We used 80% AMT and 600 pulses for our cTBS protocol following the parameters used in the original publication on cTBS by Huang et al. (2005). This cTBS protocol (600 pulses administered at 80% AMT) is thought to be inhibitory, as it reduces cortical excitability (Huang et al. 2005; Wischniewski and Schutter 2015) and the magnitude of BOLD responses for up to 50-60min after application (Hubl et al. 2008). As noted by Rossi et al. (2009) in the TMS safety guidelines, most of the published studies using cTBS have adopted 80% of AMT as the stimulation intensity following Huang et al. 2005 largely without incident (whereas a seizure was reported in a study applying cTBS at an intensity of 100% of Resting Motor Threshold (which corresponds to ~ 120% of AMT)).

Thus, in this study as well as our previous cTBS work (Lapate et al. 2017; Tambini and D'Esposito 2020; Tambini, Nee, and D'Esposito 2018) we have adopted these parameters from Huang et al. (2005). We believe that our effects are robust in part due to the large number of trials per subject & TMS site (n=440/session, obtained over 6 experimental runs; n=1320 trials/subject total), as mentioned above in the context of our reply to Reviewer 2, Q2.

3. Please provide test-retest reliability data on the task, especially as the pre to post TMS sessions cannot be contiguous given the MVPA targeting.

We thank the Reviewer for this important request. In general, performance in Go/No-Go tasks have moderate-to-high test-retest reliability. For task accuracy, previously reported test-retest Pearson's r s for similarly-designed tasks have ranged between r s = [.62 -.76] for commission errors—i.e., false alarms in No-Go trials, which is the type of error that increased following mid-LPFC TMS in our study (Bender et al. 2016: r = .62; Kertzman et al. 2008: r = .64; Weafer et al. 2013: r = .65; Hedge et al. 2017: ICC = .76; reviewed in Enkavi et al. 2019, PNAS). For completeness, we note that test-retest reliability of RTs in Go/No-Go tasks (albeit not changed by TMS in our task) have included ICCs = 0.63-0.74 (Hedge et al. 2018) and r = .88 (Kertzman et al. 2008) (Bender et al. 2016; Kertzman et al. 2008; Weafer, Baggott, and de Wit 2013; Hedge, Powell, and Sumner 2018; Enkavi et al. 2019).

Because our experiment required modifying and adapting a typically fully-blocked AGNG design into a mixed block/event-related task, we estimated this task's test-retest reliability across the different fMRI sessions we conducted.

First, we examined the test-retest reliability between the two TMS+fMRI sessions, which were conducted on average 5.6 days apart (SD = 8.2 days). We found comparable numbers to what had been previously reported in the literature: commission errors (i.e., No-Go accuracy) ranging from [r s = .65 – .68]: (TMS Session 1 vs 2 *Collapsed across emotional valences*: r = 0.689, p = 1.8e-05, 95% CI = [0.443, 0.839]; *Negative valence*: r = 0.651, p = 7.3e-05, 95% CI = [0.386, 0.817]; *Positive valence*: r = 0.688, p = 1.9e-05, 95% CI = [0.442, 0.838]). When examining the test-retest reliability between the baseline (no TMS) vs. Control (S1) fMRI sessions, which were conducted on average 98 days apart (SD = 43 days), reliability was moderate, ranging from [r s = 0.55–0.66]: (Baseline vs. Control TMS session: *Collapsed across emotional valences*: r = 0.661, p = 5.2e-05, 95% CI = [0.4, 0.822]; *Negative valence*: r = 0.596, p = 4e-04, 95% CI = [0.306 0.785]; *Positive valence*: r = 0.553, p = 0.0013, 95% CI = [0.247 0.758]).

For completeness, we also report the test-retest reliability of RTs: TMS Session 1 vs 2 *Collapsed across emotional valences* (r = 0.912, p = 9e-13, 95% CI = [0.825, 0.957]; *Negative valence*: r =

0.927, $p = 6.9e-14$, 95% CI = [0.853, 0.965]; *Positive valence*: $r = 0.876$, $p = 1.1e-10$, 95% CI = [0.756, 0.939]. Test-retest reliability for RTs between baseline (no TMS) and Control (S1) TMS sessions was similar: (*Collapsed across emotional valences*: $r = 0.818$, $p = 2e-08$, 95%CI = [0.652, 0.909]; *Positive only*: $r = 0.803$, $p = 5.6e-08$, 95% CI = [0.627, 0.901]; *Negative only*: $r = 0.823$, $p = 1.4e-08$, 95% CI = [0.661, 0.911]).

We have added this information to Supplementary Materials:

Supplementary Methods (P. 1):

AGNG task: test-retest reliability

Because our experiment required modifying and adapting typically fully-blocked AGNG task designs into a mixed block/event-related task, we estimated the test-retest reliability across the different fMRI sessions we conducted. For context, Go/No-Go tasks have typically moderate-to-high test-retest reliability. Previously reported test-retest Pearson's r s for similarly designed paradigms have ranged between r s = [.62 -.76] for commission errors (i.e., false alarms in No-Go trials; Bender et al. 2016: $r = .62$; Kertzman et al. 2008: $r = .64$; Weafer et al. 2013: $r = .65$; Hedge et al. 2017: ICC = $.76^{1-4}$; reviewed in⁵). For completeness, we note that test-retest reliability of RTs in Go/No-Go tasks (albeit not changed by TMS in our task) have included ICCs= 0.63-0.74⁴ and $r = .88^{1-5}$.

First, we examined the test-retest reliability of the task as estimated using the behavior obtained in the two TMS+fMRI sessions, which were conducted on average 5.6 days apart ($SD = 8.2$ days). For commission errors, we found comparable task reliability to what had been previously reported in the literature, ranging from [r s = .65 – .68]: (TMS Session 1 vs 2 *Collapsed across emotional valences*: $r = 0.689$, $p = 1.8e-05$, 95% CI = [0.443, 0.839]; *Negative valence*: $r = 0.651$, $p = 7.3e-05$, 95% CI = [0.386, 0.817]; *Positive valence*: $r = 0.688$, $p = 1.9e-05$, 95% CI = [0.442, 0.838]). When examining the test-retest reliability between the baseline (no TMS) vs. Control (S1) fMRI sessions, which were conducted on average 98 days apart ($SD = 43$ days), reliability was moderate, ranging from [r s = 0.55–0.66]: (Baseline vs. Control TMS session: *Collapsed across emotional valences*: $r = 0.661$, $p = 5.2e-05$, 95% CI = [0.4, 0.822]; *Negative valence*: $r = 0.596$, $p = 4e-04$, 95% CI = [0.306 0.785]; *Positive valence*: $r = 0.553$, $p = 0.0013$, 95% CI = [0.247 0.758]).

Test-retest reliability of AGNG RT data were as follows: TMS Session 1 vs 2 *Collapsed across emotional valences* ($r = 0.912$, $p = 9e-13$, 95% CI = [0.825, 0.957]; *Negative valence*: $r = 0.927$, $p = 6.9e-14$, 95% CI = [0.853, 0.965]; *Positive valence*: $r = 0.876$, $p = 1.1e-10$, 95% CI = [0.756, 0.939]). Test-retest reliability for RTs between baseline (no TMS) and Control (S1) TMS sessions were comparable: (*Collapsed across emotional valences*: $r = 0.818$, $p = 2e-08$, 95% CI = [0.652, 0.909]; *Positive only*: $r = 0.803$, $p = 5.6e-08$, 95% CI = [0.627, 0.901]; *Negative only*: $r = 0.823$, $p = 1.4e-08$, 95% CI = [0.661, 0.911]).

References

- Badre, David. 2008. "Cognitive Control, Hierarchy, and the Rostro-Caudal Organization of the Frontal Lobes." *Trends in Cognitive Sciences* 12 (5): 193–200.
- Badre, David, and Mark D'Esposito. 2009. "Is the Rostro-Caudal Axis of the Frontal Lobe Hierarchical?" *Nature Reviews. Neuroscience* 10 (9): 659–69.
- Badre, David, and Derek Evan Nee. 2018. "Frontal Cortex and the Hierarchical Control of Behavior." *Trends in Cognitive Sciences* 22 (2): 170–88.
- Bender, Angela D., Hannah L. Filmer, K. G. Garner, Claire K. Naughtin, and Paul E. Dux. 2016. "On the Relationship between Response Selection and Response Inhibition: An Individual Differences Approach." *Attention, Perception & Psychophysics* 78 (8): 2420–32.
- Bhandari, Apoorva, Christopher Gagne, and David Badre. 2018. "Just above Chance: Is It Harder to Decode Information from Prefrontal Cortex Hemodynamic Activity Patterns?" *Journal of Cognitive Neuroscience* 30 (10): 1473–98.
- Birn, R. M., A. J. Shackman, J. A. Oler, L. E. Williams, D. R. McFarlin, G. M. Rogers, S. E. Shelton, et al. 2014. "Evolutionarily Conserved Prefrontal-Amygdalar Dysfunction in Early-Life Anxiety." *Molecular Psychiatry* 19 (8): 915–22.
- Bode, Stefan, and John-Dylan Haynes. 2009. "Decoding Sequential Stages of Task Preparation in the Human Brain." *NeuroImage* 45 (2): 606–13.
- Bramson, Bob, Davide Folloni, Lennart Verhagen, Bart Hartogsveld, Rogier B. Mars, Ivan Toni, and Karin Roelofs. 2020. "Human Lateral Frontal Pole Contributes to Control over Emotional Approach–Avoidance Actions." *The Journal of Neuroscience: The Official Journal of the Society for Neuroscience* 40 (14): 2925–34.
- Bramson, Bob, Ivan Toni, and Karin Roelofs. 2023. "Emotion Regulation from an Action-Control Perspective." *Neuroscience and Biobehavioral Reviews* 153 (October): 105397.
- Cash, Robin F. H., Luca Cocchi, Jinglei Lv, Yumeng Wu, Paul B. Fitzgerald, and Andrew Zalesky. 2021. "Personalized Connectivity-Guided DLPFC-TMS for Depression: Advancing Computational Feasibility, Precision and Reproducibility." *Human Brain Mapping* 42 (13): 4155–72.
- Cash, Robin F. H., Anne Weigand, Andrew Zalesky, Shan H. Siddiqi, Jonathan Downar, Paul B. Fitzgerald, and Michael D. Fox. 2021. "Using Brain Imaging to Improve Spatial Targeting of Transcranial Magnetic Stimulation for Depression." *Biological Psychiatry* 90 (10): 689–700.
- Cole, Michael W., Takuya Ito, and Todd S. Braver. 2016. "The Behavioral Relevance of Task Information in Human Prefrontal Cortex." *Cerebral Cortex* 26 (6): 2497–2505.
- Coutanche, Marc N., Sarah H. Solomon, and Sharon L. Thompson-Schill. 2016. "A Meta-Analysis of fMRI Decoding: Quantifying Influences on Human Visual Population Codes." *Neuropsychologia* 82 (February): 134–41.
- Dixon, Matthew L., Alejandro De La Vega, Caitlin Mills, Jessica Andrews-Hanna, R. Nathan Spreng, Michael W. Cole, and Kalina Christoff. 2018. "Heterogeneity within the Frontoparietal Control Network and Its Relationship to the Default and Dorsal Attention Networks." *Proceedings of the National Academy of Sciences of the United States of America* 115 (7): E1598–1607.
- Duncan, John. 2010. "The Multiple-Demand (MD) System of the Primate Brain: Mental Programs for Intelligent Behaviour." *Trends in Cognitive Sciences* 14 (4): 172–79.
- Enkavi, A. Zeynep, Ian W. Eisenberg, Patrick G. Bissett, Gina L. Mazza, David P. MacKinnon, Lisa A. Marsch, and Russell A. Poldrack. 2019. "Large-Scale Analysis of Test–Retest Reliabilities of Self-Regulation Measures." *Proceedings of the National Academy of Sciences* 116 (12): 5472–77.
- Erickson, Kristine, Wayne C. Drevets, Luke Clark, Dara M. Cannon, Earle E. Bain, Carlos A. Zarate, Dennis S. Charney, and Barbara J. Sahakian. 2005. "Mood-Congruent Bias in Affective Go/No-Go Performance of Unmedicated Patients With Major Depressive Disorder." *American Journal of Psychiatry* 162 (11): 2171–73.
- Fischl, Bruce, Niranjini Rajendran, Evelina Busa, Jean Augustinack, Oliver Hinds, B. T. Thomas Yeo, Hartmut Mohlberg, Katrin Amunts, and Karl Zilles. 2008. "Cortical Folding Patterns and Predicting Cytoarchitecture." *Cerebral Cortex* 18 (8): 1973–80.

- Folloni, Davide, Jerome Sallet, Alexandre A. Khrapitchev, Nicola Sibson, Lennart Verhagen, and Rogier B. Mars. 2019. "Dichotomous Organization of Amygdala/Temporal-Prefrontal Bundles in Both Humans and Monkeys." *ELife* 8 (November): e47175.
- Freedberg, Michael, Jack A. Reeves, Andrew C. Toader, Molly S. Hermiller, Joel L. Voss, and Eric M. Wassermann. 2019. "Persistent Enhancement of Hippocampal Network Connectivity by Parietal RTMS Is Reproducible." *ENeuro* 6 (5). <https://doi.org/10.1523/ENEURO.0129-19.2019>.
- Freund, Michael C., Joset A. Etzel, and Todd S. Braver. 2021. "Neural Coding of Cognitive Control: The Representational Similarity Analysis Approach." *Trends in Cognitive Sciences*, April. <https://www.ncbi.nlm.nih.gov/pubmed/33895065>.
- Fuster, J. M. 2001. "The Prefrontal Cortex--an Update: Time Is of the Essence." *Neuron* 30 (2): 319–33.
- Gordon, Evan M., Timothy O. Laumann, Babatunde Adeyemo, and Steven E. Petersen. 2015. "Individual Variability of the System-Level Organization of the Human Brain." *Cerebral Cortex* 27 (1): 386–99.
- Groppa, S., A. Oliviero, A. Eisen, A. Quartarone, L. G. Cohen, V. Mall, A. Kaelin-Lang, et al. 2012. "A Practical Guide to Diagnostic Transcranial Magnetic Stimulation: Report of an IFCN Committee." *Clinical Neurophysiology: Official Journal of the International Federation of Clinical Neurophysiology* 123 (5): 858–82.
- Hamidi, Massihullah, Heleen A. Slagter, Giulio Tononi, and Bradley R. Postle. 2009. "Repetitive Transcranial Magnetic Stimulation Affects Behavior by Biasing Endogenous Cortical Oscillations." *Frontiers in Integrative Neuroscience* 3 (June): 14.
- Hare, Todd A., Colin F. Camerer, and Antonio Rangel. 2009. "Self-Control in Decision-Making Involves Modulation of the VmPFC Valuation System." *Science* 324 (5927): 646–48.
- Harfmann, Elisabeth J., Kathleen T. Rhyner, and Rick E. Ingram. 2019. "Cognitive Inhibition and Attentional Biases in the Affective Go/No-Go Performance of Depressed, Suicidal Populations." *Journal of Affective Disorders* 256 (September): 228–33.
- Haufe, Stefan, Frank Meinecke, Kai Gorgen, Sven Dahne, John-Dylan Haynes, Benjamin Blankertz, and Felix Biemann. 2014. "On the Interpretation of Weight Vectors of Linear Models in Multivariate Neuroimaging." *NeuroImage* 87 (February): 96–110.
- Hedge, Craig, Georgina Powell, and Petroc Sumner. 2018. "The Reliability Paradox: Why Robust Cognitive Tasks Do Not Produce Reliable Individual Differences." *Behavior Research Methods* 50 (3): 1166–86.
- Huang, Ying-Zu, Mark J. Edwards, Elisabeth Rounis, Kailash P. Bhatia, and John C. Rothwell. 2005. "Theta Burst Stimulation of the Human Motor Cortex." *Neuron* 45 (2): 201–6.
- Hubl, D., T. Nyffeler, P. Wurtz, S. Chaves, T. Pflugshaupt, M. Luthi, R. von Wartburg, et al. 2008. "Time Course of Blood Oxygenation Level-Dependent Signal Response after Theta Burst Transcranial Magnetic Stimulation of the Frontal Eye Field." *Neuroscience* 151 (3): 921–28.
- Jackson, Jade B., Eva Feredoes, Anina N. Rich, Michael Lindner, and Alexandra Woolgar. 2021. "Concurrent Neuroimaging and Neurostimulation Reveals a Causal Role for DIPFC in Coding of Task-Relevant Information." *Communications Biology* 4 (1): 588.
- Kaldewaij, Reinoud, Saskia B. J. Koch, Mahur M. Hashemi, Wei Zhang, Floris Klumpers, and Karin Roelofs. 2021. "Anterior Prefrontal Brain Activity during Emotion Control Predicts Resilience to Post-Traumatic Stress Symptoms." *Nature Human Behaviour*, February. <https://doi.org/10.1038/s41562-021-01055-2>.
- Kaldewaij, Reinoud, Saskia B. J. Koch, Wei Zhang, Mahur M. Hashemi, Floris Klumpers, and Karin Roelofs. 2019. "Frontal Control Over Automatic Emotional Action Tendencies Predicts Acute Stress Responsivity." *Biological Psychiatry: Cognitive Neuroscience and Neuroimaging* 4 (11): 975–83.
- Kenwood, Margaux M., Ned H. Kalin, and Helen Barbas. 2022. "The Prefrontal Cortex, Pathological Anxiety, and Anxiety Disorders." *Neuropsychopharmacology: Official Publication of the American College of Neuropsychopharmacology* 47 (1): 260–75.
- Kertzman, Semion, Katherine Lowengrub, Anat Aizer, Michael Vainder, Moshe Kotler, and Pinhas N. Dannon. 2008. "Go-No-Go Performance in Pathological Gamblers." *Psychiatry Research* 161 (1): 1–10.
- Kilford, Emma J., Lucy Foulkes, Robert Potter, Stephan Collishaw, Anita Thapar, and Frances Rice. 2015. "Affective Bias and Current, Past and Future Adolescent Depression: A Familial High Risk Study." *Journal of Affective Disorders* 174 (March): 265–71.
- Klooster, Deborah C. W., Michael A. Ferguson, Paul A. J. M. Boon, and Chris Baeken. 2022. "Personalizing Repetitive Transcranial Magnetic Stimulation Parameters for Depression Treatment Using

- Multimodal Neuroimaging." *Biological Psychiatry. Cognitive Neuroscience and Neuroimaging* 7 (6): 536–45.
- Knight, and D'Esposito. 2003. "11 Lateral Prefrontal Syndrome: A Disorder of Executive Control." *Neurological Foundations of Cognitive*. http://lncdc.free.fr/Ressources/bio_books/neurosciences/Neurological%20Foundations%20of%20Cognitive%20Neuroscience%20-%20Mark%20D'Esposito.pdf#page=272.
- Koch, Saskia B. J., Rogier B. Mars, Ivan Toni, and Karin Roelofs. 2018. "Emotional Control, Reappraised." *Neuroscience and Biobehavioral Reviews* 95 (December): 528–34.
- Koechlin, Etienne, Chrystèle Ody, and Frédérique Kounieher. 2003. "The Architecture of Cognitive Control in the Human Prefrontal Cortex." *Science* 302 (5648): 1181–85.
- Koechlin, Etienne, and Christopher Summerfield. 2007. "An Information Theoretical Approach to Prefrontal Executive Function." *Trends in Cognitive Sciences* 11 (6): 229–35.
- Lapate, Regina C., Ian C. Ballard, Marisa K. Heckner, and Mark D'Esposito. 2022. "Emotional Context Sculpts Action Goal Representations in the Lateral Frontal Pole." *The Journal of Neuroscience: The Official Journal of the Society for Neuroscience* 42 (8): 1529–41.
- Lapate, Regina C., Jason Samaha, Bas Rokers, Hamdi Hamzah, Bradley R. Postle, and Richard J. Davidson. 2017. "Inhibition of Lateral Prefrontal Cortex Produces Emotionally Biased First Impressions: A Transcranial Magnetic Stimulation and Electroencephalography Study." *Psychological Science* 28 (7): 942–53.
- Lapate, Regina C., Jason Samaha, Bas Rokers, Bradley R. Postle, and Richard J. Davidson. 2020. "Perceptual Metacognition of Human Faces Is Causally Supported by Function of the Lateral Prefrontal Cortex." *Communications Biology* 3 (1): 1–10.
- Mante, Valerio, David Sussillo, Krishna V. Shenoy, and William T. Newsome. 2013. "Context-Dependent Computation by Recurrent Dynamics in Prefrontal Cortex." *Nature* 503 (7474): 78–84.
- Menon, Vinod, and Mark D'Esposito. 2022. "The Role of PFC Networks in Cognitive Control and Executive Function." *Neuropsychopharmacology: Official Publication of the American College of Neuropsychopharmacology* 47 (1): 90–103.
- Miller, E. K., and J. D. Cohen. 2001. "An Integrative Theory of Prefrontal Cortex Function." *Annual Review of Neuroscience* 24: 167–202.
- Miller, Jacob A., Willa I. Voorhies, Daniel J. Lurie, Mark D'Esposito, and Kevin S. Weiner. 2021. "Overlooked Tertiary Sulci Serve as a Meso-Scale Link between Microstructural and Functional Properties of Human Lateral Prefrontal Cortex." *The Journal of Neuroscience: The Official Journal of the Society for Neuroscience* 41 (10): 2229–44.
- Morawetz, Carmen, Stefan Bode, Juergen Baudewig, Arthur M. Jacobs, and Hauke R. Heekeren. 2016. "Neural Representation of Emotion Regulation Goals." *Human Brain Mapping* 37 (2): 600–620.
- Mueller, Sophia, Danhong Wang, Michael D. Fox, B. T. Thomas Yeo, Jorge Sepulcre, Mert R. Sabuncu, Rebecca Shafee, Jie Lu, and Hesheng Liu. 2013. "Individual Variability in Functional Connectivity Architecture of the Human Brain." *Neuron* 77 (3): 586–95.
- Myruski, Sarah, George A. Bonanno, Olga Gulyayeva, Laura J. Egan, and Tracy A. Dennis-Tiway. 2017. "Neurocognitive Assessment of Emotional Context Sensitivity." *Cognitive, Affective & Behavioral Neuroscience* 17 (5): 1058–71.
- Nee, Derek Evan. 2021. "Integrative Frontal-Parietal Dynamics Supporting Cognitive Control." *ELife* 10 (March). <https://www.ncbi.nlm.nih.gov/pubmed/33650966>.
- Nee, Derek Evan, and Mark D'Esposito. 2016. "The Hierarchical Organization of the Lateral Prefrontal Cortex." *ELife* 5 (March). <https://www.ncbi.nlm.nih.gov/pubmed/26999822>.
- . 2017. "Causal Evidence for Lateral Prefrontal Cortex Dynamics Supporting Cognitive Control." *ELife* 6 (September): e28040.
- Nelissen, Natalie, Mark Stokes, Anna C. Nobre, and Matthew F. S. Rushworth. 2013. "Frontal and Parietal Cortical Interactions with Distributed Visual Representations during Selective Attention and Action Selection." *The Journal of Neuroscience: The Official Journal of the Society for Neuroscience* 33 (42): 16443–58.
- Paquola, Casey, Reinder Vos De Wael, Konrad Wagstyl, Richard A. I. Bethlehem, Seok-Jun Hong, Jakob Seidlitz, Edward T. Bullmore, et al. 2019. "Microstructural and Functional Gradients Are Increasingly Dissociated in Transmodal Cortices." *PLoS Biology* 17 (5): e3000284.
- Peters, Jan, and Mark D'Esposito. 2016. "Effects of Medial Orbitofrontal Cortex Lesions on Self-Control in Intertemporal Choice." *Current Biology: CB* 26 (19): 2625–28.

- Petro, Nathan M., Tien T. Tong, Daniel J. Henley, and Mital Neta. 2018. "Individual Differences in Valence Bias: fMRI Evidence of the Initial Negativity Hypothesis." *Social Cognitive and Affective Neuroscience* 13 (7): 687–98.
- Pitts, Mckinney, and Derek Evan Nee. 2022. "Generalizing the Control Architecture of the Lateral Prefrontal Cortex." *Neurobiology of Learning and Memory* 195 (November): 107688.
- Pizzagalli, Diego A., and Angela C. Roberts. 2022. "Prefrontal Cortex and Depression." *Neuropsychopharmacology: Official Publication of the American College of Neuropsychopharmacology* 47 (1): 225–46.
- Polanía, Rafael, Michael A. Nitsche, and Christian C. Ruff. 2018. "Studying and Modifying Brain Function with Non-Invasive Brain Stimulation." *Nature Neuroscience* 21 (2): 174–87.
- Rajkowska, G., and P. S. Goldman-Rakic. 1995. "Cytoarchitectonic Definition of Prefrontal Areas in the Normal Human Cortex: II. Variability in Locations of Areas 9 and 46 and Relationship to the Talairach Coordinate System." *Cerebral Cortex* 5 (4): 323–37.
- Rigotti, Mattia, Omri Barak, Melissa R. Warden, Xiao-Jing Wang, Nathaniel D. Daw, Earl K. Miller, and Stefano Fusi. 2013. "The Importance of Mixed Selectivity in Complex Cognitive Tasks." *Nature* 497 (7451): 585–90.
- Roelofs, Karin, Bob Bramson, and Ivan Toni. 2023. "A Neurocognitive Theory of Flexible Emotion Control: The Role of the Lateral Frontal Pole in Emotion Regulation." *Annals of the New York Academy of Sciences* 1525 (1): 28–40.
- Rose, Nathan S., Joshua J. LaRocque, Adam C. Riggall, Olivia Gosseries, Michael J. Starrett, Emma E. Meyering, and Bradley R. Postle. 2016. "Reactivation of Latent Working Memories with Transcranial Magnetic Stimulation." *Science* 354 (6316): 1136–39.
- Rossi, Simone, Mark Hallett, Paolo M. Rossini, Alvaro Pascual-Leone, and Safety of TMS Consensus Group. 2009. "Safety, Ethical Considerations, and Application Guidelines for the Use of Transcranial Magnetic Stimulation in Clinical Practice and Research." *Clinical Neurophysiology: Official Journal of the International Federation of Clinical Neurophysiology* 120 (12): 2008–39.
- Sack, Alexander T., Roi Cohen Kadosh, Teresa Schuhmann, Michelle Moerel, Vincent Walsh, and Rainer Goebel. 2009. "Optimizing Functional Accuracy of TMS in Cognitive Studies: A Comparison of Methods." *Journal of Cognitive Neuroscience* 21 (2): 207–21.
- Sakai, Katsuyuki, and Richard E. Passingham. 2006. "Prefrontal Set Activity Predicts Rule-Specific Neural Processing during Subsequent Cognitive Performance." *The Journal of Neuroscience: The Official Journal of the Society for Neuroscience* 26 (4): 1211–18.
- Satpute, A. B., and K. A. Lindquist. 2019. "The Default Mode Network's Role in Discrete Emotion." *Trends in Cognitive Sciences*. https://www.sciencedirect.com/science/article/pii/S1364661319301780?casa_token=4gecT7rdKHwAAAAA:MePVBp2cOAWMfO2S_IO3iKJW8-EPEew83DNz_Pjgc2YAAxEO-tOYHO2kjCyR1NfWDI6iAXxNY5U.
- Stokes, Mark G., Christopher D. Chambers, Ian C. Gould, Tracy R. Henderson, Natasha E. Janko, Nicholas B. Allen, and Jason B. Mattingley. 2005. "Simple Metric For Scaling Motor Threshold Based on Scalp-Cortex Distance: Application to Studies Using Transcranial Magnetic Stimulation." *Journal of Neurophysiology* 94 (6): 4520–27.
- Tambini, Arielle, and Mark D'Esposito. 2020. "Causal Contribution of Awake Post-Encoding Processes to Episodic Memory Consolidation." *Current Biology: CB* 30 (18): 3533-3543.e7.
- Tambini, Arielle, Derek Evan Nee, and Mark D'Esposito. 2018. "Hippocampal-Targeted Theta-Burst Stimulation Enhances Associative Memory Formation." *Journal of Cognitive Neuroscience* 30 (10): 1452–72.
- Todd, Michael T., Leigh E. Nystrom, and Jonathan D. Cohen. 2013. "Confounds in Multivariate Pattern Analysis: Theory and Rule Representation Case Study." *NeuroImage* 77 (August): 157–65.
- Vázquez-Rodríguez, Bertha, Laura E. Suárez, Ross D. Markello, Golia Shafiei, Casey Paquola, Patric Hagmann, Martijn P. van den Heuvel, Boris C. Bernhardt, R. Nathan Spreng, and Bratislav Misic. 2019. "Gradients of Structure–Function Tethering across Neocortex." *Proceedings of the National Academy of Sciences* 116 (42): 21219–27.
- Wang, Jane X., Lynn M. Rogers, Evan Z. Gross, Anthony J. Ryals, Mehmet E. Dokucu, Kelly L. Brandstatt, Molly S. Hermiller, and Joel L. Voss. 2014. "Targeted Enhancement of Cortical-Hippocampal Brain Networks and Associative Memory." *Science* 345 (6200): 1054–57.

- Waskom, Michael L., Dharshan Kumaran, Alan M. Gordon, Jesse Rissman, and Anthony D. Wagner. 2014. "Frontoparietal Representations of Task Context Support the Flexible Control of Goal-Directed Cognition." *The Journal of Neuroscience: The Official Journal of the Society for Neuroscience* 34 (32): 10743–55.
- Weafer, Jessica, Matthew J. Baggott, and Harriet de Wit. 2013. "Test-Retest Reliability of Behavioral Measures of Impulsive Choice, Impulsive Action, and Inattention." *Experimental and Clinical Psychopharmacology* 21 (6): 475–81.
- Westin, Gregory G., Bruce D. Bassi, Sarah H. Lisanby, and Bruce Luber. 2014. "Determination of Motor Threshold Using Visual Observation Overestimates Transcranial Magnetic Stimulation Dosage: Safety Implications." *Clinical Neurophysiology: Official Journal of the International Federation of Clinical Neurophysiology* 125 (1): 142–47.
- Wischnewski, Miles, and Dennis J. L. G. Schutter. 2015. "Efficacy and Time Course of Theta Burst Stimulation in Healthy Humans." *Brain Stimulation* 8 (4): 685–92.
- Woolgar, Alexandra, Russell Thompson, Daniel Bor, and John Duncan. 2011. "Multi-Voxel Coding of Stimuli, Rules, and Responses in Human Frontoparietal Cortex." *NeuroImage* 56 (2): 744–52.
- Yeo, B. T. Thomas, Fenna M. Krienen, Jorge Sepulcre, Mert R. Sabuncu, Danial Lashkari, Marisa Hollinshead, Joshua L. Roffman, et al. 2011. "The Organization of the Human Cerebral Cortex Estimated by Intrinsic Functional Connectivity." *Journal of Neurophysiology* 106 (3): 1125–65.

REVIEWER COMMENTS

Reviewer #1 (Remarks to the Author):

My concerns have been addressed

Reviewer #2 (Remarks to the Author):

The authors have done a thorough job responding to all concerns raised in the last round of review. They have not, however, adequately addressed them.

Regarding the issue of low decoding accuracies, observing consistently weak but "above chance" decoding performance across subjects (e.g., many subjects with accuracies ranging from 51% to 53% accuracy) may suggest that there is a meaningful (e.g., medium to large) effect in the population. The newly reported effect sizes show this, more or less. Observing a large effect at the group level does not suggest that action-goal states can be accurately decoded within any individual. Because AUCs in the range of .533 are typically considered "poor", I remain unconvinced that the information being decoded is related to action-goal states per se as opposed to some other variable.

Given that univariate and multivariate effects are very close spatially (presumably at or near the spatial precision of TMS), I am increasingly convinced that different stimulation procedures would have produced similar effects.

I remain unconvinced that claims such as "In summary, individualized, information-guided cTBS targeting task-relevant action goal signals in mid-LPFC robustly reduced the strength of mid-LPFC action-goal representations" are supported by the data.

No new data are presented to suggest that individualized targeting contributed to the effectiveness of TMS.

No new data are presented to suggest that information-guided targeting contributed to the effectiveness of TMS (if anything the overlap with univariate effects suggests they would not make much of a difference).

No new data are presented to suggest that within-subject decoders reflect action-goal states as opposed to some other information that very weakly differentiates task conditions.

REVIEWER COMMENTS

Reviewer #1 (Remarks to the Author):

My concerns have been addressed.

Reviewer #2 (Remarks to the Author):

The authors have done a thorough job responding to all concerns raised in the last round of review. They have not, however, adequately addressed them.

R2.1.1. Regarding the issue of low decoding accuracies, observing consistently weak but above chance decoding performance across subjects (e.g., many subjects with accuracies ranging from 51% to 53% accuracy) may suggest that there is a meaningful (e.g., medium to large) effect in the population. The newly reported effect sizes show this, more or less. Observing a large effect at the group level does not suggest that action-goal states can be accurately decoded within any individual.

Indeed, to recapitulate, action-goal decoding in mid-LPFC at baseline and in the Control TMS session had reliable and moderate effect sizes at the group level: Cohen's $d = 0.707$ and $d = 0.655$, respectively.

In the previous revision, this Reviewer suggested that we quantified the number of subjects showing reductions in mid-LPFC action-goal decoding following cTBS to mid-LPFC (which were: 21/31 subjects (67.74%) relative to the baseline session and 20/31 subjects (64.52%) relative to the Control S1 session; reported & included in the manuscript as part of **Revision 1**);

We have now taken a similar approach to examine action-decoding data at the individual-subject level for the *non*-mid-LPFC sessions—i.e., at baseline (no TMS) and following TMS to the active Control (S1) site. Here, we report (a) the number of subjects showing AUC > 0.5 for action-goal decoding in those sessions, as well as (b) the number of subjects showing action-goal decoding that is statistically significantly above chance at the individual-subject and TMS-session level (cross-validated classifier performance that was greater than 95% of the shuffled data from that TMS+fMRI session; $n=500$ permutations). We report these results using an unbiased, anatomical mid-LPFC ROI. We found that 22/31 (71%) subjects had action-goal AUC > 0.5 in the baseline fMRI session (10/31 were significantly above chance at the single subject/session level; i.e., classifier performance fell at > 95% of the shuffled data distribution), and 22/31 (71%) had action-goal AUC > 0.5 in the Control fMRI session (15/31 were statistically above chance at the single subject/session level). See **Figure R1**.

Figure R1. (A & B) Subject-level classifier performance (AUC) for action-goal decoding (Go/No-Go) in mid-LPFC observed in **(A)** the baseline (no TMS) fMRI session and **(B)** the TMS Control (S1) fMRI session.

We have added these data and figures to the manuscript as *Supplementary Figure 1A-B* (Pages 2-3 of *Supplementary Results*), which are referred to in the Results section (P. 6)

“(...) for individual-level results, see *Supplementary Results and Figure S1*.”

R2.1.2. Because AUCs in the range of .533 are typically considered "poor", I remain unconvinced that the information being decoded is related to action-goal states per se as opposed to some other variable.

As mentioned in our prior Revision, the base rate of multivariate decoding in PFC is lower than in other brain regions. A recent meta-analysis of $n=76$ studies indicate that our decoding accuracies are within the expected range for this region. As reviewed by Bhandari et al. 2018, decoding accuracies in PFC for two-way classifications often hover between 53-55% [e.g. (Nelissen et al. 2013; Woolgar et al. 2011; Bode and Haynes 2009)]—consistent with the mean $AUC = .533$ decoding accuracy obtained in the baseline fMRI session in the current study. That is believed to be in part due to weaker spatial clustering of neurons in mid-LPFC compared to other regions (Bhandari, Gagne, and Badre 2018). Therefore, studies conducting multivariate decoding in prefrontal cortex must employ a high number of trials to detect reliable (even if numerically ‘small’) effects, which is what we did in this study ($n=440$ /session, obtained over 6 experimental runs; $n=1320$ trials/subject total).

We have conducted the following analyses to establish specificity of the decoding to action-goal states (vs. other plausible task factors): (1) Examined whether one can decode emotional valence (the other main task manipulated variable) from mid-LPFC, and whether cTBS to mid-LPFC changed emotional valence decoding; (2) **(New data analysis.)** Examined whether action-goal decoding in mid-LPFC (and its reduction by mid-LPFC cTBS) remains reliable after excluding neutral-face trials (which were present in the No-Go condition).

Finally, we (3) include below (and in the revised manuscript) a discussion of the different aspects of *action-goal states* that may be decoded from mid-LPFC and modulated by TMS, as well as associated interpretational limitations, which we hope speaks to the Reviewer’s concerns.

- (1) **Representational specificity (Action Goals vs. Emotional Valence):** Mid-LPFC decoding of action goals was not driven by emotional valence. Moreover, mid-LPFC cTBS (targeting mid-LPFC based on the individualized location of action goals) did not modulate the decoding of emotion valence, but was specific to the decoding of action-goals.

Emotional valence (Positive vs. Negative cues) was not decodable above chance from mid-LPFC voxels at baseline (no TMS) (AUC $M = 0.502$, $B = 0.002$, $SE = 0.011$, $t = 0.178$, Cohen's $d = 0.028$, $p = 0.86$) (**Figure S6**). Consistent with the lack of reliable emotional valence decoding at baseline, following cTBS, decoding of emotional valence remained at chance in mid-LPFC (cTBS to mid-LPFC: AUC $M = 0.498$ $SE = 0.009$, $t = -0.241$, Cohen's $d = 0.052$, $p = 0.811$; cTBS to Control site: AUC $M = 0.49$, $SE = 0.01$, $t = -1.295$, Cohen's $d = 0.138$, $p = 0.205$), with no change as a function of TMS site (cTBS * region interaction: $F = 1.419$, $p = 0.243$). Further demonstrating the informational specificity of our cTBS approach targeting mid-LPFC action-goals, the cTBS-driven reduction in classifier evidence in mid-LPFC was significantly stronger for action goals (Figure 2A) than for emotional valence (Figure S6), as indicated by a significant cTBS*information type (action vs. valence) interaction in mid-LPFC voxels ($F = 4.207$, $p = 0.015$).

Figure S6. Multivariate classifier performance for decoding of emotional valence (Positive vs. Negative) in individualized mid-LPFC TMS sites is plotted (AUC) as a function of TMS condition. In contrast to action-goal evidence (Figure 2A), classifier performance was at chance for emotional valence decoding at baseline, and unchanged by cTBS to mid-LPFC.

- (2) **(New data analysis.) To further establish whether decoding was driven by action-goal representations, as opposed to other, potentially confounding, task variables,** we re-analyzed the strength of action-goal decoding in mid-LPFC sites (both at baseline as well as their change after mid-LPFC TMS) after excluding an emotional-valence task factor that covaried with Go vs. No-Go classes: neutral-face trials, which were present in No-Go trials only. This directly speaks to the reviewer's concern, as these trial types could have been responsible for partially driving classification since they were correlated with the task conditions being decoded. As shown in **Figure R2**, our originally reported results (Figure 2A) were fully replicated after excluding neutral-face trials.

Figure R2. Multivariate classifier performance for decoding of action goals (Go vs. No-Go AUC, *excluding neutral-face trials*) in individualized LPFC TMS sites is plotted as a function of TMS condition. In agreement with results shown in Fig. 2A, action-goal decoding is reliable at baseline (no TMS) ($M = 0.536$, $t = 2.753$, $p_{vs\ chance} = 0.0138$) and following cTBS to the active TMS Control site ($M = 0.525$, $t = 2.274$, $p_{vs\ chance} = 0.0299$). Following cTBS to mid-LPFC, action-goal decoding in mid-LPFC dropped to chance ($M = 0.494$, $t = -0.486$, $p_{vs\ chance} = 0.63$) and was significantly reduced compared to both baseline and Control (S1) TMS sessions ($p = 0.0015$ and $p = 0.0194$,

respectively; cTBS site main effect: $F = 5.48$, $p = 0.0044$.)

(3) Discussion of action goal states: interpretational limitations of the current study.

We acknowledge that action-goal states exist in a continuum of abstract to concrete, which may be part of what the Reviewer is alluding to (re: which variable is being decoded)—a fully abstract action-goal versus an action plan coupled with execution. Isolating the precise aspect(s) of action-goal states that are being decoded and changed by TMS would require de-coupling Go vs. No-Go abstract goals from their associated action by including an additional task manipulation/condition, which cannot be achieved with the current dataset. Note that additional data collection is not feasible: a dataset of this type (longitudinal, within-subjects TMS + fMRI) takes ~1.5 year(s) to collect and analyze, and the lead author has since moved institutions. We had previously alluded to this interpretational limitation in the Discussion section (see below). In response to the Reviewer’s concern, we now more explicitly acknowledge this point, which we agree constitutes an important direction for future work:

Discussion, P. 11-12:

In the future, it will be important to establish whether the nature and function of mid-LPFC goal-relevant representations as revealed by tasks with explicitly cued goals (such as the AGNG task employed in the current study) generalize to spontaneously initiated behavioral goals in less structured scenarios, including abstract goals less coupled to action (motoric) components. Of note, cTBS to mid-LPFC (vs. Control) did not significantly change response times (**Figure S3**), suggesting that mid-LPFC action-goal representations targeted with cTBS likely comprised relatively abstract goals that went beyond mere motor preparation and/or execution. **Nonetheless, the current dataset does not address whether action-goal states decodable in this region in our study, and altered by mid-LPFC cTBS, pertained to fully abstract action-goal representations versus motoric action plans.** Therefore, future studies employing naturalistic paradigms—including self-

relevant, ecologically valid stimuli—that capture behavioral goals varying in abstraction, combined with representational analysis approaches (Freund, Etzel, and Braver 2021), will be required to fully unveil the nature and format of LPFC control signals that modulate emotional behavior.

R2.2. Given that univariate and multivariate effects are very close spatially (presumably at or near the spatial precision of TMS), I am increasingly convinced that different stimulation procedures would have produced similar effects.

We believe this is unlikely for a number of reasons, listed below. (But we also acknowledge that one cannot entirely rule out this possibility without collecting a third and/or fourth TMS session + fMRI scanning day *per participant* for an explicit comparison—which is not feasible with the current dataset.)

- (1) The Reviewer's original question about the location of univariate vs. multivariate effects pertained to the 'center of mass' in the mid-LPFC ROI—i.e., in practice, the average location of individualized, information-based ROIs vs. the average location of the maximal univariate activations within mid-LPFC. However, we did not target an 'average mid-LPFC location' in this study—rather, we targeted individualized mid-LPFC locations that were widely distributed across mid-LPFC. Therefore, the distance between *average* multivariate vs. *average* univariate effects is not directly relevant for interpreting the current results. We have now also computed the absolute distance between *individualized* univariate peaks and individualized multivariate, information-based sites, which more directly speaks to how targeting would have differed, had it been based off univariate sites. These distances are as follows (in *mm*): For Go vs. No-Go ($x = 11.419, y = 12.710, z = 16.129$); for No-Go vs. Go ($x = 8, y = 15.355, z = 15.742$). In summary, the average distance between univariate *versus* multivariate peaks across (x, y, z) axes is 1.3 cm.
- - Individualized LPFC ROIs
- (2) Even if the distance between average locations were relevant for the stimulation protocol we adopted, 1cm+ apart is above the spatial resolution of TMS for figure-of-eight coils (0.5-1cm) (Sliwinska, Vitello, and Devlin 2014; Toschi et al. 2008; Romanella et al. 2020). Therefore, had we targeted the average univariate (versus the average multivariate) location in mid-LPFC, TMS effects would a-priori be expected to differ given the spatial resolution of this technique (and likewise had we targeted individualized univariate sites).
 - (3) Critically, directly underscoring the dissociation of univariate vs. multivariate effects in our data—and the specificity of TMS effects to multivariate, and not univariate—results, cTBS to mid-LPFC did *not* alter univariate responses in mid-LPFC, only multivariate, action-goal signals. (See **Figure S7** below.)
 - (4) Relatedly, and further addressed in response to Reviewer point R2.3 (below), the robust reduction of action-goal decoding in mid-LPFC was specific to individualized, functionally defined ROIs, and not present in the larger, anatomical mid-LPFC ROI. Therefore, this strongly suggests that an ROI-specific stimulation procedure produced ROI-specific (i.e., regionally specific) results.

Figure S7. Univariate mid-LPFC activity is plotted as a function of Action Goal (Go vs. No-Go) and TMS Condition (no TMS, Control/S1, and mid-LPFC). Information-guided cTBS to mid-LPFC reduced multivariate decoding of action goals (Go vs. No-Go; **Figure 2A**) without altering overall univariate activity in mid-LPFC (cTBS site main effect n.s. $F = 0.92$, $p = 0.409$). The interaction of action goal (Go vs. No-Go) * cTBS site on mid-LPFC univariate activity was also non-significant ($F = 0.69$, $p = 0.506$), thereby underscoring the specificity of the impact of information-guided TMS to multivariate (vs. univariate) action-goal signals.

R2.3. I remain unconvinced that claims such as "In summary, individualized, information-guided cTBS targeting task-relevant action goal signals in mid-LPFC robustly reduced the strength of mid-LPFC action-goal representations" are supported by the data.

No new data are presented to suggest that individualized targeting contributed to the effectiveness of TMS.

No new data are presented to suggest that information-guided targeting contributed to the effectiveness of TMS (if anything the overlap with univariate effects suggests they would not make much of a difference).

We thank the Reviewer for inviting us to further clarify the specificity of individualized, information-based mid-LPFC TMS to individualized ROIs (and to information-based metrics) in our manuscript, as well as to acknowledge interpretational limitations of the current study.

Firstly, as detailed in our prior Revision (**Revision 1**), we used an *individualized* TMS approach because prior work has suggested its increased efficacy—compared to group-based coordinates (Sack et al. 2009; Klooster et al. 2022; Cash, Cocchi, et al. 2021; Cash, Weigand, et al. 2021). Given those data suggesting that individualized TMS targeting is more effective, and previous work demonstrating the feasibility of using a multivariate (*information-based*) approach (Rose et al. 2016), we chose to adopt that strategy in this study with the goal of maximizing statistical power and sensitivity (which is an ethical consideration in TMS studies). This was mentioned in the paper in Response to this Reviewer's previous comment (Revision 1):

Introduction (Page 5):

This individualized TMS targeting approach follows prior work (Rose et al. 2016; Polanía, Nitsche, and Ruff 2018) and aimed to maximize the functional specificity and sensitivity of TMS (Sack et al. 2009; Cash, Cocchi, et al. 2021; Polanía, Nitsche, and Ruff 2018) in a region characterized by large inter-individual variability in anatomy (Miller et al. 2021; Rajkowska and Goldman-Rakic 1995; Gordon et al. 2015; Fischl et al. 2008) as well as

in anatomy-function correspondence (Paquola et al. 2019; Vázquez-Rodríguez et al. 2019; Mueller et al. 2013).

That said, as mentioned in our prior Revision, we agree with the Reviewer that methodological work systematically comparing the efficacy of individualized, multivariate-based TMS sites vs. group-coordinate based (or univariate based) TMS targeting will be important in the future, as direct comparisons between distinct TMS targeting approaches (in the same individuals) are rare. That is also an inherent limitation of the current work: we show robust effects of an individualized, MVPA-based TMS targeting approach—compared to baseline and an active control TMS site (S1)—as opposed to compared to another TMS condition that targets the same region (mid-LPFC) based on a *non-individualized* (e.g., group) coordinate (which would be required to establish the ‘superiority’ of one targeting the strategy over another). Doing the latter was not the goal of the present study, and would have required additional TMS+fMRI sessions systematically varying the targeting strategy within the same region in the same individuals, which is not feasible in the current dataset and would have likely caused high subject attrition given the large number of TMS+fMRI sessions required. Instead, our inferences, made relative to an active Control TMS site (and baseline), are bolstered by a number of control analyses establishing the regional and representational specificity of our results (listed below).

We previously acknowledged this methodological point in the Supplementary Discussion. In response to the Reviewer’s comments (Revision 2), we have now expanded this Discussion, contextualized it within the broader literature of individualized brain stimulation in psychiatry, explicitly mentioned the interpretational limitation(s) of different control site strategies, and moved it to the main manuscript for increased transparency, as follows:

Pages 12-13:

Relatedly, a methodological innovation of the current work was to target mid-LPFC based on the location of individualized action-goal (Go vs. No-Go) multivariate representations, following recent empirical and theoretical work on combined brain stimulation and neuroimaging (Rose et al. 2016; Polanía, Nitsche, and Ruff 2018) and a prior study showing the increased efficacy of individualized, fMRI-based TMS targeting (Sack et al. 2009). Consistent with our core hypothesis, the strength of action-goal decoding in individualized mid-LPFC sites was reduced following cTBS to mid-LPFC (compared to baseline and Control (S1) sessions), an effect that was not observed in anatomical and/or non-individualized mid-LPFC sites (**Figures S8-9**). Collectively, these results suggest the import of parsing inter-individual heterogeneity in mid-LPFC function (Miller et al. 2021; Rajkowska and Goldman-Rakic 1995; Gordon et al. 2015; Fischl et al. 2008; Paquola et al. 2019; Vázquez-Rodríguez et al. 2019; Mueller et al. 2013) and offer support for an individualized, functional-based approach to brain stimulation aimed at understanding

causality in brain-behavior relationships (Rose et al. 2016; Polanía, Nitsche, and Ruff 2018; Sack et al. 2009), in alignment with a growing emphasis on precision neuroimaging (Gordon et al. 2017; Braga and Buckner 2017) and precision psychiatry for brain stimulation (Roalf, Figeo, and Oathes 2024; Cash et al. 2019). Early work systematically comparing individualized vs. group-based TMS targeting approaches demonstrated the increased potency of individualized, fMRI-guided TMS for altering behavior (Sack et al. 2009), and recent studies of TMS as a treatment for depression have embraced individualized, fMRI-based targeting, serving as the basis for the first FDA-approved individualized fMRI-based TMS protocol for depression treatment (Williams 2021; Cole et al. 2022; Roalf, Figeo, and Oathes 2024). Note however that the current study did not directly test whether individualized targeting of multivariate representations *per se* was required for the observed findings, as they were examined relative to the active control TMS site (S1) and baseline sessions (rather than relative to a group-coordinate based mid-LPFC TMS site). Therefore, future work systematically quantifying the impact of distinct individualized vs. group based TMS-targeting strategies (guided by univariate, multivariate, and/or functional-connectivity-based signals)—ideally conducted within subjects—will be required to more fully characterize their differential efficacy in modulating neural activity and behavior, and inform translational neuroscience efforts, including personalized brain stimulation strategies that are increasingly embraced in the clinic (Roalf, Figeo, and Oathes 2024; Cash, Cocchi, et al. 2021; Cash et al. 2019).

Secondly, we previously conducted a number of control analyses to establish the functional (*regional* and *representational*) specificity of our findings, but those data were in the Supplementary Material and might have been overlooked.

Therefore, we now display them here—in combination with **new analyses** we conducted in response to Reviewer 2's new comments, which we believe directly speak to the Reviewer's concern:

- (1) **Individualized ROI specificity (new analysis of data in Fig. S8): Mid-LPFC action-goal decoding changes after mid-LPFC cTBS are specific to individualized (functionally defined, and not anatomically defined) mid-LPFC targets**

As mentioned, the strength of action-goal decoding in individualized mid-LPFC sites was reduced following cTBS to mid-LPFC (compared to both baseline and Control (S1) sessions, $p_s < 0.0023$, **Figure 2A** included for reference). To examine whether this effect was specific to individualized, functionally defined mid-LPFC targets targeted by TMS, we tested whether cTBS to mid-LPFC reduced action-goal representations in the anatomically defined mid-LPFC ROI.

We found that cTBS to mid-LPFC did *not* change the strength of action-goal representations when an anatomically defined mid-LPFC mask was examined (**Figure S8**): in this anatomical ROI, action-goal decoding remained significant following cTBS to mid-LPFC ($M = 0.545$, $B = 0.045$ ($SE = 0.01$), $t = 4.47$, Cohen's $d = 0.764$, $p < 0.001$) and was comparable to decoding observed in both baseline (no TMS) and TMS Control (S1) sessions ($M = 0.533$, $B = 0.033$ ($SE = 0.01$), $t = 3.215$, Cohen's $d = 0.707$, $p = 0.007$ and $M = 0.545$, $B = 0.045$ ($SE = 0.012$), $t = 3.874$, Cohen's $d = 0.655$, $p = 0.001$, respectively). Accordingly, the main effect of cTBS site was non-significant in anatomically defined mid-LPFC ($p > 0.173$), which contrasts starkly with results obtained when examining functionally specific, individualized cTBS sites (**Figure 2A & Figure S2**).

Figure S8. Multivariate classifier performance (decoding of action goal: Go vs. No-Go) from an anatomically defined mid-LPFC region is plotted (AUC) as a function of TMS condition. Following cTBS to mid-LPFC, action-goal remained decodable above chance from an anatomically defined mid-LPFC mask, in contrast with results observed when probing individual-specific and functionally defined mid-LPFC sites (Figure 2A). ** $p < 0.01$

Following the Reviewer's input (Revision 2), we have now formally tested the interaction of ROI * TMS condition (ROI: anatomical vs. individualized (functional) * TMS condition: baseline, Control vs. mid-LPFC), which examines whether the modulation of action-goal decoding in mid-LPFC by TMS was specific to individualized (vs. anatomically defined) mid-LPFC targets. We found that this interaction was indeed significant, ($F = 5.733$, $p = 0.003$), indicating that action-goal decoding in mid-LPFC was significantly disrupted by TMS only in individualized, functionally defined ROIs, but not in the larger, anatomical mid-LPFC ROI (mid-LPFC AUC pairwise ROI comparison: $t = 4.466$, $p < 0.0001$).

(This new analysis has been added to: 'Regional specificity of MVPA changes by mid-LPFC cTBS: anatomical vs. individualized mid-LPFC ROIs' under **Control Analyses** in Supplementary Results).

- (2) **New analysis:** Mid-LPFC action-goal decoding changes after mid-LPFC cTBS are specific to individualized mid-LPFC ROIs compared to functionally (information-based) defined, but not-individualized, mid-LPFC ROIs

Following the Reviewer's ongoing concern (Revision 2), we conducted an entirely new analyses of this dataset to test whether there was evidence that action-goal decoding in subject-specific, individualized mid-LPFC ROIs (functionally defined based on action-goal decoding from their baseline fMRI session, and targeted by TMS) was any different than action-goal decoding in *non-individualized*, but equivalently-sized and functionally-defined mid-LPFC ROIs obtained from *other subjects*. To do so, for each subject, we extracted and averaged action-goal classifier AUC by TMS condition from *all other subjects'* mid-LPFC ROIs that did not spatially overlap with that subject's own individualized mid-LPFC ROI. In other words, going a step beyond the aforementioned analyses using a large, anatomically defined mid-LPFC ROI (Fig. S8), we now probe *other subjects'* functionally defined, individualized ROIs as the control site. These new analyses permitted us to address the following two critical points:

- (A) Are subject-specific mid-LPFC ROIs stable? If so, one would expect action-goal decoding (AUC) to be significantly *stronger* in individualized (compared to non-individualized) mid-LPFC in an independent session (here, the Control TMS session).
- (B) Is the reduction in mid-LPFC action-goal decoding, produced by targeting individualized, information-based mid-LPFC ROIs, *more robust* in the targeted, individualized (compared to non-individualized) mid-LPFC ROIs? If so, one would expect a significant interaction between the magnitude of mid-LPFC cTBS effects and whether the mid-LPFC ROI examined was individualized. In other words, putative *inhibition* by individualized mid-LPFC cTBS targeting, *if specific*, should be manifested as significantly *lower* classifier action-goal decoding (AUC) in individualized mid-LPFC ROIs than in non-individualized mid-LPFC ROIs.

As shown in **Figure R3**, both critical predictions were confirmed by these new analyses. First, indicating that individualized (information-based) ROIs were stable and better captured action-goal information in mid-LPFC (compared to non-individualized mid-LPFC ROIs), action-goal decoding in mid-LPFC obtained in an independent session—the Control/S1 session—was *stronger* in individualized ROIs compared to non-individualized mid-LPFC ROIs ($p = 0.045$). Second and critically, whether the mid-LPFC ROI was individualized significantly modulated the main effect of TMS condition, as indicated by a strong ROI * TMS condition interaction, $F = 9.226$, $p = 0.00011$. Specifically, the significant *reduction* of classifier action-goal decoding in mid-LPFC by cTBS was specific to individualized (targeted) mid-LPFC ROIs compared to in non-individualized ROI. As originally reported in the manuscript, following cTBS to mid-LPFC, action-goal decoding (AUC) in subject's individualized (i.e., targeted) mid-LPFC ROIs dropped to chance levels ($p_{vs.chance} > 0.885$), and was significantly reduced compared to both baseline (no TMS) ($p = 0.0006$) and Control (S1) TMS sessions ($p = 0.0007$), TMS main effect $F = 7.117$, $p = 0.0009$. Notably, the main effect of TMS condition was absent when examining equally sized, but non-individualized LPFC ROIs (TMS main effect $F = 1.617$, $p > 0.214$). Action-goal decoding in non-individualized (non-targeted) mid-LPFC ROIs remained above chance following mid-LPFC cTBS ($p_{vs.chance} = 0.004$) and was not reduced compared to either baseline (no TMS) ($p > 0.9405$) or Control (S1) TMS sessions ($p > 0.398$). Moreover, as expected, action goal decoding following mid-LPFC (putatively inhibitory) cTBS was significantly *lower* in individualized mid-LPFC ROIs targeted by TMS compared to non-individualized mid-LPFC ROIs ($p = 0.039$) (**Figure R3**).

In summary, these analyses (1) underscore the stability of action-goal decoding in subject-specific mid-LPFC ROIs; and (2) provide evidence of the regional specificity of the results of individualized, information-based targeting to subject's individualized TMS targets, relative to *both* (a) an anatomically defined mid-LPFC (**Figure S8**); and critically, (b) functionally defined, equally sized and equivalently functionally defined non-individualized mid-LPFC ROIs (i.e., using individualized mid-LPFC ROI from *other subjects* as the comparison site; **Figure R3**).

These new data and Figure R3 have been added to the manuscript as ‘*Regional specificity of MVPA changes by mid-LPFC cTBS: individualized vs. non-individualized (but functionally defined) mid-LPFC ROIs*’ under **Control Analyses** and (new) **Figure S9** in Supplementary Results Pages 12-14).

Figure R3. Multivariate classifier performance in LPFC as a function of whether the LPFC ROI examined was individualized (information-based) versus not. Multivariate classifier performance of action-goal decoding (Go vs. No-Go classifier AUC) from mid-LPFC sites is plotted as a function of TMS condition and whether the location of the LPFC ROI examined was individualized based on subject’s location of peak classifier performance at the baseline fMRI session (no TMS) (originally reported analyses; Individualized ROI: ‘Yes’) versus from *other subjects’* LPFC ROIs (mean AUC) (Individualized ROI: ‘No’). Whether the mid-LPFC ROI examined was individualized significantly modulated the main effect of TMS condition, as indicated by a significant ROI * TMS condition interaction, $F = 9.226$, $p = 0.00011$. As originally reported in the manuscript, following cTBS to mid-LPFC, action-goal decoding (AUC) in subject’s individualized (and targeted by TMS) mid-LPFC ROIs dropped to chance levels ($p_{vs.chance} > 0.885$), and was significantly reduced compared to *both* baseline (no TMS) ($p = 0.0006$) and Control (S1) TMS sessions ($p = 0.0007$), TMS main effect $F = 7.117$, $p = 0.0009$. Notably, a TMS condition main effect is absent when examining equally sized, but non-individualized LPFC ROIs (i.e., average Go vs. No-Go classifier AUC obtained from other individuals), TMS main effect N.S. ($F = 1.617$, $p > 0.214$). Specifically, decoding in non-individualized mid-LPFC ROIs remained above chance ($p_{vs.chance} = 0.004$) and was *not* reduced following mid-LPFC cTBS compared to either baseline (no TMS) ($p > 0.9405$) or Control (S1) TMS sessions ($p > 0.398$). Critically, TMS significantly reduced classifier decoding in individualized (targeted) mid-LPFC ROIs compared to in non-individualized ROI ($p = 0.039$). Moreover, as would be expected if individualized (information-based) ROIs captured action-goal information in mid-LPFC better and more reliably than non-individualized ROIs, action-goal decoding in mid-LPFC obtained in an independent session—the Control/S1 session—was stronger in individualized compared to non-individualized ROIs ($p = 0.045$).

P values legend: *** $p \leq 0.001$; ** $p < .01$; * $p < .05$

Light gray: individual condition p values (action goal decoding) against chance

Medium gray: pairwise p value for individualized mid-LPFC ROI in the Control TMS condition

Blue: pairwise p value for individualized mid-LPFC ROI in the mid-LPFC TMS condition
Black: pairwise p values for TMS condition contrasts (per ROI condition)

- (3) **Representational specificity (Action Goals vs. Emotional Valence)**: Mid-LPFC decoding of action goals was not driven by emotional valence. Moreover, mid-LPFC cTBS did not modulate the decoding of emotion valence, but was specific to the decoding of action-goals

This point was articulated (and relevant **Fig. S6** shown) above in response to Reviewer's point R2.1.2.

- (4) **Representational specificity (multivariate vs. univariate)**: Mid-LPFC changes after mid-LPFC cTBS in individualized, mid-LPFC ROIs (functionally defined based on location of action-goal decoding) are specific to the decoding of action-goals (Go/No-Go) and not univariate activity.

This point was articulated (and relevant **Fig. S7** shown) above in response to Reviewer's point R2.2.

These existing and new control analyses are included in the manuscript as follows:

P. 10:

Supporting the specificity of our information-based TMS approach, cTBS targeting individualized action-goal representations in mid-LPFC did not alter emotional-valence representations—or the overall magnitude of univariate activation—in subject-specific mid-LPFC sites ($p_s > .24$) (*Supplementary Information: Control Analyses*; **Figures S6-S7**). Moreover, cTBS-induced reductions in action-goal decoding in mid-LPFC were specific to individualized mid-LPFC sites targeted by TMS, and were not observed in *non*-subject specific, anatomically defined mid-LPFC ($p > .17$) (**Figure S8**) or in *non*-subject specific, functionally-defined mid-LPFC ROIs ($p > .214$) (**Figure S9**; see *Supplementary Information: Control Analyses: Regional specificity of MVPA changes by mid-LPFC cTBS*).

R2.4. No new data are presented to suggest that within-subject decoders reflect action-goal states as opposed to some other information that very weakly differentiates task conditions.

New analysis quantifying individual level, within-subject decoding AUCs can be found above (Response R2.1.1; **Fig. R1 A-B**).

New and previous data analysis pertaining to potential task confounds that could drive mid-LPFC decoding were presented above (Response R2.1.2; **Fig. R2**; **Fig. S6**).

To summarize the study design, the Affective Go/No-Go task we used in this study orthogonally manipulated action goals and Emotional valence. Classifiers were trained on action-goal

decoding. Effect sizes for action-goal decoding in mid-LPFC were robust and consistent across baseline and Control TMS sessions: baseline session (vs. chance): Cohen's $d = 0.707$; Control (S1) TMS (vs. chance): Cohen's $d = 0.655$. cTBS targeting individualized action-goal decoding sites in mid-LPFC (vs. cTBS to an active control region (S1) and relative to baseline) decreased the strength of action goal decoding in mid-LPFC, reducing action-goal decoding in individualized mid-LPFC ROIs to chance levels, with highly consistent effect sizes relative to both baseline and the TMS Control (S1) sessions (Cohen's $d = 0.403$ and $d = 0.448$, respectively).

We performed numerous control analyses that established the specificity of these effects to multivariate decoding of action goals and to individualized (information-based, targeted by TMS) mid-LPFC ROIs, many of which are detailed above: (1) action goal decoding was unchanged by mid-LPFC cTBS in an anatomical (*non-individualized*) mid-LPFC ROI or in other subjects' equally sized, individualized ROIs; (2) emotional valence decoding in mid-LPFC was at chance and unchanged by mid-LPFC cTBS targeting action goals—thereby underscoring the representational specificity of our targeting approach; (3) univariate signals in mid-LPFC were unchanged by mid-LPFC cTBS—thereby further underscoring the specificity of our results to information-based metrics (which were used to guide TMS targeting), rather than global, univariate changes.

Critically, cTBS targeting individualized action goal decoding sites in mid-LPFC not only produced a robust reduction of mid-LPFC action goal decoding in mid-LPFC, but impacted **goal-oriented performance in the task**, reducing No-Go accuracy, particularly in negative-cue trials. These results are detailed in the manuscript.

In summary, information-guided TMS reduced mid-LPFC action-goal representations and impaired goal-directed behavior in a task requiring cognitive control in response to emotional cues. The combined TMS+fMRI approach allowed us to ascertain the functional specificity of our findings, as detailed above. Although we are unclear on what other alternative hypotheses Reviewer 2 has in mind, we include the control analyses mentioned above in the letter and manuscript, and we hope that this summary helps integrate across the different pieces of evidence that support the findings and conclusions reported in the manuscript.

References

- Bhandari, Apoorva, Christopher Gagne, and David Badre. 2018. "Just above Chance: Is It Harder to Decode Information from Prefrontal Cortex Hemodynamic Activity Patterns?" *Journal of Cognitive Neuroscience* 30 (10): 1473–98.
- Braga, Rodrigo M., and Randy L. Buckner. 2017. "Parallel Interdigitated Distributed Networks within the Individual Estimated by Intrinsic Functional Connectivity." *Neuron* 95 (2): 457–471.e5.
- Cash, Robin F. H., Luca Cocchi, Jinglei Lv, Yumeng Wu, Paul B. Fitzgerald, and Andrew Zalesky. 2021. "Personalized Connectivity-Guided DLPFC-TMS for Depression: Advancing Computational Feasibility, Precision and Reproducibility." *Human Brain Mapping* 42 (13): 4155–72.
- Cash, Robin F. H., Anne Weigand, Andrew Zalesky, Shan H. Siddiqi, Jonathan Downar, Paul B. Fitzgerald, and Michael D. Fox. 2021. "Using Brain Imaging to Improve Spatial Targeting of Transcranial Magnetic Stimulation for Depression." *Biological Psychiatry* 90 (10): 689–700.
- Cash, Robin F. H., Andrew Zalesky, Richard H. Thomson, Ye Tian, Luca Cocchi, and Paul B. Fitzgerald. 2019. "Subgenual Functional Connectivity Predicts Antidepressant Treatment Response to Transcranial Magnetic Stimulation: Independent Validation and Evaluation of Personalization." *Biological Psychiatry* 86 (2): e5–7.
- Cole, Eleanor J., Angela L. Phillips, Brandon S. Bentzley, Katy H. Stimpson, Romina Nejad, Fahim Barmak, Clive Veerapal, et al. 2022. "Stanford Neuromodulation Therapy (SNT): A Double-Blind Randomized Controlled Trial." *American Journal of Psychiatry* 179 (2): 132–41.
- Fischl, Bruce, Niranjini Rajendran, Evelina Busa, Jean Augustinack, Oliver Hinds, B. T. Thomas Yeo, Hartmut Mohlberg, Katrin Amunts, and Karl Zilles. 2008. "Cortical Folding Patterns and Predicting Cytoarchitecture." *Cerebral Cortex* 18 (8): 1973–80.
- Freund, Michael C., Joset A. Etzel, and Todd S. Braver. 2021. "Neural Coding of Cognitive Control: The Representational Similarity Analysis Approach." *Trends in Cognitive Sciences*, April. <https://www.ncbi.nlm.nih.gov/pubmed/33895065>.
- Gordon, Evan M., Timothy O. Laumann, Babatunde Adeyemo, and Steven E. Petersen. 2015. "Individual Variability of the System-Level Organization of the Human Brain." *Cerebral Cortex* 27 (1): 386–99.
- Gordon, Evan M., Timothy O. Laumann, Adrian W. Gilmore, Dillan J. Newbold, Deanna J. Greene, Jeffrey J. Berg, Mario Ortega, et al. 2017. "Precision Functional Mapping of Individual Human Brains." *Neuron* 95 (4): 791–807.e7.
- Klooster, Deborah C. W., Michael A. Ferguson, Paul A. J. M. Boon, and Chris Baeken. 2022. "Personalizing Repetitive Transcranial Magnetic Stimulation Parameters for Depression Treatment Using Multimodal Neuroimaging." *Biological Psychiatry. Cognitive Neuroscience and Neuroimaging* 7 (6): 536–45.
- Miller, Jacob A., Willa I. Voorhies, Daniel J. Lurie, Mark D'Esposito, and Kevin S. Weiner. 2021. "Overlooked Tertiary Sulci Serve as a Meso-Scale Link between Microstructural and Functional Properties of Human Lateral Prefrontal Cortex." *The Journal of Neuroscience: The Official Journal of the Society for Neuroscience* 41 (10): 2229–44.
- Mueller, Sophia, Danhong Wang, Michael D. Fox, B. T. Thomas Yeo, Jorge Sepulcre, Mert R. Sabuncu, Rebecca Shafee, Jie Lu, and Hesheng Liu. 2013. "Individual Variability in Functional Connectivity Architecture of the Human Brain." *Neuron* 77 (3): 586–95.
- Paquola, Casey, Reinder Vos De Wael, Konrad Wagstyl, Richard A. I. Bethlehem, Seok-Jun Hong, Jakob Seidlitz, Edward T. Bullmore, et al. 2019. "Microstructural and Functional Gradients Are Increasingly Dissociated in Transmodal Cortices." *PLoS Biology* 17 (5): e3000284.

- Polanía, Rafael, Michael A. Nitsche, and Christian C. Ruff. 2018. "Studying and Modifying Brain Function with Non-Invasive Brain Stimulation." *Nature Neuroscience* 21 (2): 174–87.
- Rajkowska, G., and P. S. Goldman-Rakic. 1995. "Cytoarchitectonic Definition of Prefrontal Areas in the Normal Human Cortex: II. Variability in Locations of Areas 9 and 46 and Relationship to the Talairach Coordinate System." *Cerebral Cortex* 5 (4): 323–37.
- Roalf, David R., Martijn Figee, and Desmond J. Oathes. 2024. "Elevating the Field for Applying Neuroimaging to Individual Patients in Psychiatry." *Translational Psychiatry* 14 (1): 87.
- Romanella, S. M., G. Sprugnoli, G. Ruffini, K. Seyedmadani, S. Rossi, and E. Santarnecchi. 2020. "Noninvasive Brain Stimulation & Space Exploration: Opportunities and Challenges." *Neuroscience and Biobehavioral Reviews* 119 (December): 294–319.
- Rose, Nathan S., Joshua J. LaRocque, Adam C. Riggall, Olivia Gosseries, Michael J. Starrett, Emma E. Meyering, and Bradley R. Postle. 2016. "Reactivation of Latent Working Memories with Transcranial Magnetic Stimulation." *Science* 354 (6316): 1136–39.
- Sack, Alexander T., Roi Cohen Kadosh, Teresa Schuhmann, Michelle Moerel, Vincent Walsh, and Rainer Goebel. 2009. "Optimizing Functional Accuracy of TMS in Cognitive Studies: A Comparison of Methods." *Journal of Cognitive Neuroscience* 21 (2): 207–21.
- Toschi, Nicola, Tobias Welt, Maria Guerrisi, and Martin E. Keck. 2008. "A Reconstruction of the Conductive Phenomena Elicited by Transcranial Magnetic Stimulation in Heterogeneous Brain Tissue." *Physica Medica: PM: An International Journal Devoted to the Applications of Physics to Medicine and Biology: Official Journal of the Italian Association of Biomedical Physics* 24 (2): 80–86.
- Vázquez-Rodríguez, Bertha, Laura E. Suárez, Ross D. Markello, Golia Shafiei, Casey Paquola, Patric Hagmann, Martijn P. van den Heuvel, Boris C. Bernhardt, R. Nathan Spreng, and Bratislav Misic. 2019. "Gradients of Structure–Function Tethering across Neocortex." *Proceedings of the National Academy of Sciences* 116 (42): 21219–27.
- Williams, Nolan. 2021. "Stanford Neuromodulation Therapy (SNT): A Double-Blinded, Randomized, and Controlled Trial." *Brain Stimulation: Basic, Translational, and Clinical Research in Neuromodulation* 14 (6): 1736–37.

REVIEWERS' COMMENTS

Reviewer #2 (Remarks to the Author):

The authors have satisfactorily addressed my concerns.

REVIEWERS' COMMENTS

Reviewer #2 (Remarks to the Author):

The authors have satisfactorily addressed my concerns.

We thank Reviewer 2 for their thoughtful questions and comments.